# MEASUREMENT INFORMATION MULTIPLE-REUSE AL-LOWS DEEPER QUANTUM TRANSFORMER

## ABSTRACT

The current era has witnessed the success of the transformer in the field of classical deep neural networks (DNNs) and the potential of quantum computing. One naturally expects that quantum computing can offer significant speedup for the transformer. Recent developments of quantum transformer models are faced with challenges including the expensive cost of non-linear operations and the information loss problem caused by measurements. To address this issue, this paper proposes a scheme called measurement information multiple-reuse (MIMR). MIMR enables the repeated utilization of intermediate measurement data from former layers, thus enhancing information-transferring efficiency. This scheme facilitates our quantum vision transformer (QViT) capable of achieving exponential speedup compared to classical counterparts, with the support of many parameters and large depth. Our QViT model is further examined with an instance of 86 million parameters, which halves the requirements for tomography error compared to the one without MIMR. This demonstrates the superior performance of MIMR over existing schemes. Our findings underscore the importance of exploiting the value of information from each measurement, offering a key strategy towards scalable quantum deep neural networks.

## 1 INTRODUCTION

The transformative era of deep learning has witnessed the rise of varieties of large-scale models, wherein the transformer Vaswani et al. (2017) emerges as a cornerstone in this evolution. At the heart of the transformer's success lies its attention mechanism, a paradigm-shifting approach that allows for the effective management of billions of parameters, maintaining trainability and adaptability across diverse applications. However, the computing resource of the transformer scales quadratically with the sequence length. This limitation has emerged as a bottleneck in the continued scaling of transformer models, necessitating innovative approaches to extend their capabilities.

Quantum computing is a promising solution to the computational limitations of classical machine learning, offering exponential speedup and enhanced computing capabilities. In the field of quantum deep neural networks (QDNNs), plenty of works have been proposed Beer et al. (2020); Liu et al. (2024); Li et al. (2020); Kerenidis et al. (2020a). Their potential applications include fields of image recognition Li et al. (2020), quantum physics Liu et al. (2022), data classification Hur et al. (2022), and so forth. Despite the relatively mature development of QDNNs, the advancement of quantum algorithms for transformers has lagged behind.

Early attempts at quantum transformer have been made, based on either variational quantum circuits (VQCs) Cerezo et al. (2021), or quantum linear algebra (QLA) Childs et al. (2017); Liu et al. (2021); Krovi (2023). The VQC-based quantum transformers Cherrat et al. (2022); Evans et al. (2024) lack provable quantum advantage and also suffer from trainability problems like barren plateaus Wang et al. (2021); McClean et al. (2018) and local minima Anschuetz & Kiani (2022); Bittel & Kliesch (2021). The QLA-based quantum transformers Guo et al. (2024); Liao & Ferrie (2024) have theoretical speedup while lacking full end-to-end implementation. Neither multi-layer implementation nor backpropagation has been realized yet. Nikhil et al. presented a quantum transformer model which utilizes the Linear Combination of Unitaries and Quantum Singular Value Transform primitives as building blocks, this model provides a quantum attention layer, the implementation of other layers,

the complexity of quantum-classical data conversion between different layers, and the end-to-end implementation needs further investigation.

In this work, we propose a full implementation of a multi-layer quantum transformer based on QLA, including the realization of forward pass and backpropagation. Our quantum transformer has exponential speedup on the sequence length on both forward pass and backpropagation compared to the classical counterpart. To enable the stacking of layers, a major improvement of our work is the utilization strategy of classical information reuse. While the classical information that can be extracted from the quantum state in every single measurement is limited, we argue that the measured information has not been carefully utilized in previous works, inducing an unexpected information loss and forbidding the deepening of a quantum neural network. This phenomenon has been observed in the similar QLA-based quantum deep neural network Kerenidis et al. (2020a) and is further explored in this work. To address this issue, we propose the Measurement Information Multiple-Reuse (MIMR) scheme to mitigate the information loss across layers by making full use of the measured information. To showcase the utility of MIMR, we construct a quantum vision transformer (QViT) with 86 million parameters, demonstrating improvement in accuracy with image classification tasks of real-world datasets as well as strong robustness against information loss.

## 2 MULTIPLE-REUSE OF MEASUREMENT INFORMATION

### 2.1 MOTIVATION: NECESSITY OF INFORMATION REUSE IN CONSTRUCTING QUANTUM DEEP NEURAL NETWORK

In this section, we discuss the pivotal role of reusing intermediate measurement information to accelerate classical deep neural networks (DNNs) via quantum computing. A multi-layer DNN, involving nonlinear operations at each layer, can be viewed as a discrete nonlinear system where each layer symbolizes a step in its evolution. Current research indicates that quantum computing struggles to effectively accelerate the evolution of strongly nonlinear systems, with complexity potentially increasing exponentially with the number of evolution steps Liu et al. (2021). This exponential increase in complexity directly conflicts with the expectations for quantum speedup and presents a significant challenge for QDNNs. This discussion extends to limitations in efficiently implementing quantum backpropagation, necessitated not only by the non-linear operations of gradients but also by the need for information on intermediate quantum states.

One approach to addressing the challenges mentioned above involves incorporating measurement operations at intermediate steps of QDNNs. Specifically, we can introduce intermediate measurements after each layer, using measurement outcomes to reconstruct the output before feeding it into the next layer. Consequently, the complexity increases linearly with the number of layers. Similar ideas are employed in many recent works, including the quantum convolutional neural networks (QCNNs) Kerenidis et al. (2020a) and quantum algorithms for solving nonlinear systems Xue et al. (2021); Krovi (2023); Chen et al. (2022).

However, the cost associated with intermediate measurements can be substantial, potentially undermining the quantum advantage offered by QDNNs. Thus, a cost-effective scheme for intermediate measurements becomes essential. Intermediate measurement, a form of quantum tomography, involves several efficient tomography techniques such as $l_\infty$ tomography Kerenidis et al. (2020b), shadow tomography Aaronson (2018); Huang et al. (2020), and neural network-based methods Carrasquilla et al. (2019); Torlai et al. (2018).

Given Holevo's bound Holevo (1973), each measurement on an $n$-dimensional quantum state can extract only $\log n$ bits of information, rendering the process of extracting classical information from quantum states highly inefficient. Efficient tomography algorithms aim to reduce the number of measurements, maximizing the utility of the classical data derived from each measurement. Despite these advancements, existing quantum acceleration methods for DNNs, such as QCNNs, do not fully exploit the potential of intermediate measurements. Typically, each layer in these networks blocks the output obtained from previous layers and only conveys the most recent measurement information to the next layer.

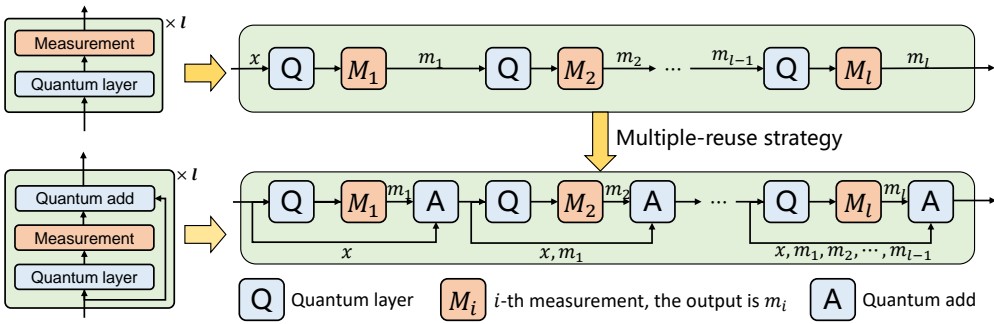

Figure 1: **Multiple-reuse strategy.**

## 2.2 MEASUREMENT INFORMATION MULTIPLE-REUSE

To address this inefficiency, we propose a multiple-reuse strategy for intermediate measurement data, developing techniques to repeatedly leverage this information throughout the network. This strategy ensures more efficient use of quantum resources and enhances the practical scalability of QDNNs.

The multiple-reuse strategy is derived from residual connection, a basic building block in DNNs. Residual connection is written as $y = x + f(x)$, where $f(x)$ is the output of a layer with input $x$. The idea of implementing a multiple-reuse strategy is to add intermediate measurements to a branch of the residual connection layer. Then the previous information can be transferred from the other branch and perform Quantum Add, as shown in Figure 1.

As a comparison, previous works adopted single-reuse strategy Kerenidis et al. (2024); Guo et al. (2024), of which the intermediate measurements are executed after each quantum layer, and only the most recent measurement results are passed to the next layer. Quantum residual connections have been explored in the context of quantum neural networks based on variational quantum circuits. Wen et al. introduced a residual connection framework utilizing the linear combination of unitary operations to enhance the expressivity of quantum neural networks on NISQ devices Wen et al. (2024). Similarly, Muhammad et al. proposed a residual approach, termed ResQNets, to mitigate the barren plateaus problem in quantum neural networks Kashif & Al-Kuwari (2024). While these residual connection strategies are well-suited for variational quantum circuits operating on NISQ devices, their direct application to quantum transformers presents significant challenges. Notably, the success rate decreases exponentially with the number of residual layers, and the computational complexity grows polynomially with the context length, posing substantial scalability limitations.

In our multiple-reuse strategy, each layer contains a residual connection that allows all previous intermediate measurement results to be passed through an independent branch of the residual connection without additional measurement. As shown in the bottom part of Figure 1, the input $x$ and the measurement results $m_1, m_2, \cdots, m_l$ are all passed to the final quantum layer. Compared to the single-reuse strategy, the data flow in the multiple-reuse strategy is more active. Wen et al. realized residual connections by the frame of a linear combination of unitary to enhance the expressivity of quantum neural networks in NISQ devices Wen et al. (2024), the complexity of their scheme increases exponentially with the residual connection layers, and their scheme cannot solve the "exponential increase" problem caused by nonlinear layers. Muhammad et al. also proposed a residual approach for mitigating barren plateaus in quantum neural networksKashif & Al-Kuwari (2024).

We can make full use of the information obtained from all intermediate measurements to build more efficient QDNNs based on the multiple-reuse strategy. To study the practical influences, in Appendix B we compare the extent of information loss using single- and multiple-reuse strategies, by observing the cosine similarity between input and output through two processes. It shows that it mitigates information loss for state reconstruction. In Figure fig-tomography-error-cub-forward, numerical evidence shows that it achieves less information loss during the forward pass of the quantum transformer, thus likely improving the effectiveness of the training process. The multiple-reuse strategy can alleviate the problem of information loss with the number of layers and therefore help build up a deeper quantum neural network.

## 3 QUANTUM VISION TRANSFORMER

In this section, we introduce our developed QViT. Traditional transformer architectures consist of nonlinear layers and non-unitary matrix computations, presenting substantial challenges in leveraging quantum computing for acceleration. Current research faces the following challenges: (1) The complexity increases exponentially with the number of layers; (2) The success probability of implementing non-unitary matrix operations in quantum linear algebra is less than 1. Our QViT effectively addresses the challenge of exponential resource growth associated with an increase in layers and significantly reduces the probabilistic steps during the execution process. This enhancement is derived from two major innovations in our implementation of QViT: (1) We employ the MIMR scheme within the transformer encoder layer to prevent the issue of complexity from increasing exponentially with the number of layers. (2) Apart from the Attention layer, the computational processes in other layers of the transformer are independent for different tokens. We implement these layers using quantum arithmetic operations, thereby circumventing the probabilistic issues typically encountered with quantum linear algebra.

Based on the above ideas, we build the complete forward pass and backpropagation process for the QViT. Both the forward pass and backpropagation processes achieve exponential speedup with respect to the sequence length $n$, while the complexity increases linearly with the number of layers $l$, as detailed in Theorem 3.1.

**Theorem 3.1 (Query complexity of forward pass and backpropagation of QViT)** *Given an input $X \in \mathbb{R}^{d \times n}$, there exists a quantum algorithm to realize the forward pass and backpropagation of an $l$ layers vision transformer, the query complexity to $X$ is $\widetilde{O}\left(\frac{ld^2 \text{polylog}(n)}{\epsilon \delta^2}\right)$, where $\delta$ represents the tomography error, and $\epsilon$ is the computational accuracy.*

**Remark.** Our QViT utilizes the MIMR scheme, incorporating $l_\infty$ tomography as described in Kerenidis et al. (2020b) within this framework. In this context, $\delta$ denotes the tomography error associated with the $l_\infty$ tomography. Detailed explanations and mathematical formulations of $l_\infty$ tomography are presented in Theorem C.2. For intuitive comparison, the complexity of the classical ViT is $\widetilde{O}(lnd(n+d)\log(1/\epsilon))$, which confirms our statement of exponential speedup with respect to $n$. The dependence on $d$ is the same as the classical ViT. $\epsilon$ and $\delta$ also influence the complexity of the QViT. The computing accuracy $\epsilon$ can be moderate in large models, such as 8-bit, or even 4-bit computing accuracy. The tomography error $\delta$ can also be moderate, in the following numerical tests, the QViT shows high performance with moderate tomography error (i.e. $\delta = 0.003$). Thus, as $n$ increases, the exponential acceleration capabilities of the QViT become increasingly pronounced.

### 3.1 FRAMEWORK

We first introduce the framework of the QViT, which is shown in Figure 2. Note that in the remaining part of the paper, we will use abbreviations to avoid repeats, see Table 4. As shown in subfigure (a), (b), and (c), the naming and usage of the major components remain the same as their classical counterparts, including QPos layer, quantum transformer encoder, and QHead layer.

There are two types of layers in the QViT, the quantum layers and the quantum-classical data transfer layers, as shown in Figure 2(d). Quantum layers are compatible with quantum input and output, providing quantum speedup with existing quantum algorithms, displayed as black circles. Quantum-classical data transfer layers, including quantum-to-classical and classical-to-quantum, displayed as red and blue circles, are used to implement the QSave and QLoad techniques, which will be further introduced in the following sections.

#### 3.1.1 QUANTUM LAYERS

Quantum layers are the layers that provide quantum speedup, including QPos, QNorm, QAttn, QAdd, and QFFN. One method to implement these operations involves quantum linear algebra, inherently producing probabilistic outcomes at each step. We notice that, apart from the QAttn layer, other layers can be regarded as $n$ independent $d$-dimensional operations. Consequently, we can implement these layers using $d$-dimensional quantum arithmetic operations, which are not affected by success probability. This approach significantly reduces the number of probabilistic steps involved.

Figure 2: **Framework of quantum vision transformer.** **(a)** The primary structure of the QViT proposed in this paper, includes a quantum version of position embedding, transformer encoder, and QHead. **(b-c)** Detailed implementations of the QHead, quantum transformer encoder, and quantum embedding layer, respectively. **(d)** The color of the logo indicates the type of each layer. **(e)** QSave & QLoad process.

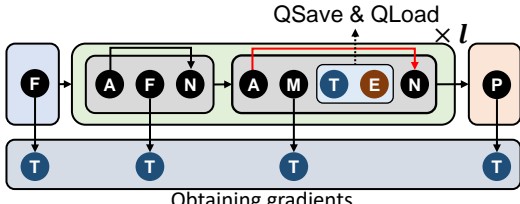

Figure 3: **Backpropagation process of QViT.**

Next, we introduce the implementation of QAttn and other layers. The QViT incorporates two distinct data encoding strategies: Analog-Encoding (A-Encoding) and Digital-Encoding (D-Encoding), defined as:

$$\text{A-Encoding} : O_A(\alpha)|0\rangle = |\alpha\rangle = \frac{1}{\|x\|}\sum_{i=0}^{n-1}\alpha_i|i\rangle, \tag{1}$$

$$\text{D-Encoding} : O_D(\alpha)|i\rangle|0\rangle = |i\rangle|\alpha_i\rangle, i = 0, 1, \cdots, n-1. \tag{2}$$

For a specific layer utilizing A-Encoding, the input/output is the amplitude encoding state of the target data. Conversely, for layers employing D-Encoding, the input/output corresponds to the $O_D$ operation on the target vector.

The input of the QAttn layer is the D-Encoding of $X$. We implement QAttn using quantum linear algebra, quantum amplitude estimation, and other algorithms to obtain the A-Encoding of the QAttn output $Y$. The details are introduced in Appendix D.4. Following the QAttn layer, quantum-classical data transfer layers are executed to achieve the D-Encoding of $\tilde{Y}$, where $\tilde{Y}$ is the sampled version of QAttn output $Y$. Detailed information about these quantum-classical data transfer layers can be found in Section 3.1.2.

Then, we establish the D-Encoding for the input to the next QAttn layer using quantum arithmetic. The implementation details are introduced in Appendix D.

### 3.1.2 QUANTUM-CLASSICAL DATA TRANSFER LAYERS

As stated in the previous section, the QViT construction process also requires intermediate measurement steps, which are realized in quantum-classical data transfer layers. The quantum-classical data transfer layers contain QSave and QLoad processes. QSave is a quantum tomography process, we use $l_\infty$ tomography to realize QSave procedure. QLoad procedure is used to encode the tomography results into the following quantum operations, the QLoad is realized by querying QRAM followed by QDAC. See Appendix C.2, C.5 and C.3 for relevant knowledge of $l_\infty$ tomography, qRAM and QDAC.

In the QViT, we add the QSave and QLoad procedures after the quantum multi-attention layer. There are two reasons: (1) The QAdd layer is behind the quantum multi-attention layer, which means we add the QSave and QLoad procedures before the QAdd layer. Then, based on the multiple-reuse strategy introduced in Section 2.2, the previous measurement information can also be reused from the other branch of the QAdd (The red line of the Figure 2(c)). (2) The quantum multi-attention layer is different from other layers, Other layers can be regarded as $n$ independent $d$-dimensional operations, but the quantum multi-attention layer cannot. So other layers can be realized by implementing $d$-dimensional quantum arithmetic operations in parallel, and the implementation process does not require measurement, the implementation details are introduced in Appendix D. Therefore, we add the QSave and QLoad procedures after QAttn.

In the QViT architecture, we integrate the QSave and QLoad procedures subsequent to the quantum multi-attention layer for two primary reasons: (1) The QSave and QLoad procedures are prior to the QAdd layer, which allows for the reuse of previous measurement information from the alternate branch of the QAdd layer, as delineated in the multiple-reuse strategy discussed in Section 2.2 and illustrated by the red line in Figure 2(c). (2) The QAttn layer differs fundamentally from other layers, which typically consist of $n$ independent $d$-dimensional operations. Unlike these layers, the quantum multi-attention layer contains $n \times n$-dimensional operations. Consequently, while other layers can execute $d$-dimensional quantum arithmetic operations in parallel without intermediate measurement, the implementation of the QAttn layer is more complex, the details are shown in Appendix D. Therefore, we add the QSave and QLoad procedures after the QAttn layer.

## 3.2 FORWARD PASS

The forward pass is directly built by executing each layer according to its definition, so we will delay the overall algorithm procedure of the forward pass to the Appendix, shown in Algorithm 1. The implementation details of all layers are explicitly shown in Appendix D.

As for complexity, because each layer equipped with quantum linear algebra has provided speedup, the overall quantum speedup is thus naturally given. The proof of Theorem 3.1 is shown in Appendix E.

## 3.3 BACKPROPAGATION

Next, we detail the backpropagation process, which mirrors the structure of the forward pass. The implementation is depicted in Figure 3. There are key differences between the backpropagation and forward propagation processes: (1) We incorporate layer tomography prior to the QAttn layer, performing tomography subsequent to the backpropagation through the QAttn layer. (2) The backpropagation process involves computing gradients for parameters across the MLP, Attention, and Position Embedding layers. We prepare the amplitude encoding of these parameter gradients by querying the intermediate data from the propagation process and then apply $l_\infty$ tomography to capture the sampled parameter gradients. Comprehensive implementation details for each phase of the backpropagation are provided in Appendix D.

The backpropagation also contains QSave and QLoad procedures, which are performed after the backpropagation of the QAttn layer. Alike the forward pass, the previous measurement information during the backpropagation process can be reused from the alternate branch of the QAdd layer, thereby implementing the multiple-reuse strategy. This approach ensures that information measured during the backpropagation is efficiently utilized.

## 4 NUMERICAL TESTS

In this section, we validate the performance of the QViT utilizing the multiple-reuse strategy through numerical tests. Specifically, we conduct the following experiments: (1) We test the impact of multiple-reuse and single-reuse strategies on the output of each layer of QViT when different tomography errors are selected. (2) We also test QViT's fine-tuning process using multiple-reuse and single-reuse strategies with different tomography errors.

### 4.1 SETUP

**Datasets.** In our simulation, we test four classification datasets: CUB-200-2011 Wah et al. (2011), Cifar-10/100 Krizhevsky et al. (2009), and Oxford-IIIT Pets Parkhi et al. (2012), the details of the datasets are listed in Table 1.

Table 1: Overview of classification datasets.

| Dataset Name | Number of Categories | Image Resolution | Dataset Size |
|---|---|---|---|
| CUB-200-2011 | 200 | Varies | 11,788 |
| CIFAR-10 | 10 | 32x32 | 60,000 |
| CIFAR-100 | 100 | 32x32 | 60,000 |
| Oxford-IIIT Pets | 37 | Approx. 200x300 | 7,349 |

**Model.** We use the "ViT-Base" model in Dosovitskiy et al. (2021). The details of the model are listed in Table 2. The hidden size $D$ is the embedding dimension of one patch, and the FFN size is the dimension of the hidden layer in feedforward.

Table 2: Details of the vision transformer.

| Model | Layer | Hidden size $D$ | MLP size | Heads | Params |
|---|---|---|---|---|---|
| ViT-Base | 12 | 768 | 3072 | 12 | 86M |

**Training and Fine-tuning.** In the training process, we use the model pre-trained on the ImageNet-21k Deng et al. (2009) and transfer the model to the specific datasets with fine-tuning. In fine-tuning process, we use AdamW Loshchilov & Hutter (2019) optimizer with lr $= 0.0001$ and weight decay $= 0.05$. The batch size is $64$.

**Hardware.** The following experiments were conducted on a server equipped with Intel Xeon Gold 6230 (2.10 GHz) × 2 and NVIDIA RTX A6000, with a total running memory of 512 GB. The training time for one fine-tuning (3000 iterations) on a single NVIDIA RTX A6000 GPU is approximately 12 hours.

**Software.** Our numerical experiments utilized MMPretrain Contributors (2023), an open-source model, as the core framework. For our study, we developed a specialized quantum deep neural network toolkit, which was instrumental in implementing the forward pass and backpropagation processes of the QViT. This toolkit features a configurable QSave operator. Designed as an extension of PyTorch, it seamlessly integrates with a broad spectrum of existing toolchains, enhancing its applicability and utility in quantum deep learning research. The source code is available at `https://github.com/anonymous0618/qvit`.

**Pre-trained model.** The pre-trained QViT model was trained on the ImageNet-21k dataset, which can be downloaded from `https://mmpretrain.readthedocs.io/en/latest/papers/vision_transformer`.

### 4.2 EFFECTS OF THE MULTIPLE-REUSE STRATEGY ON FORWARD PASS

We first study the impact of the multiple-reuse strategy on the QViT forward pass. Specifically, we evaluate the cosine similarity between the output from each layer and the expected output under

conditions of multiple-reuse and single-reuse strategies. The results, illustrated in Figure 11, are obtained using various tomography errors. The findings demonstrate that when the tomography errors are consistent, the layer outputs of the QViT with the multiple-reuse strategy significantly outperform those using the single-reuse strategy. This indicates that the multiple-reuse strategy enhances the forward pass process of the QViT.

### 4.3 EFFECTS OF MULTIPLE-REUSE STRATEGY ON FINE-TUNING

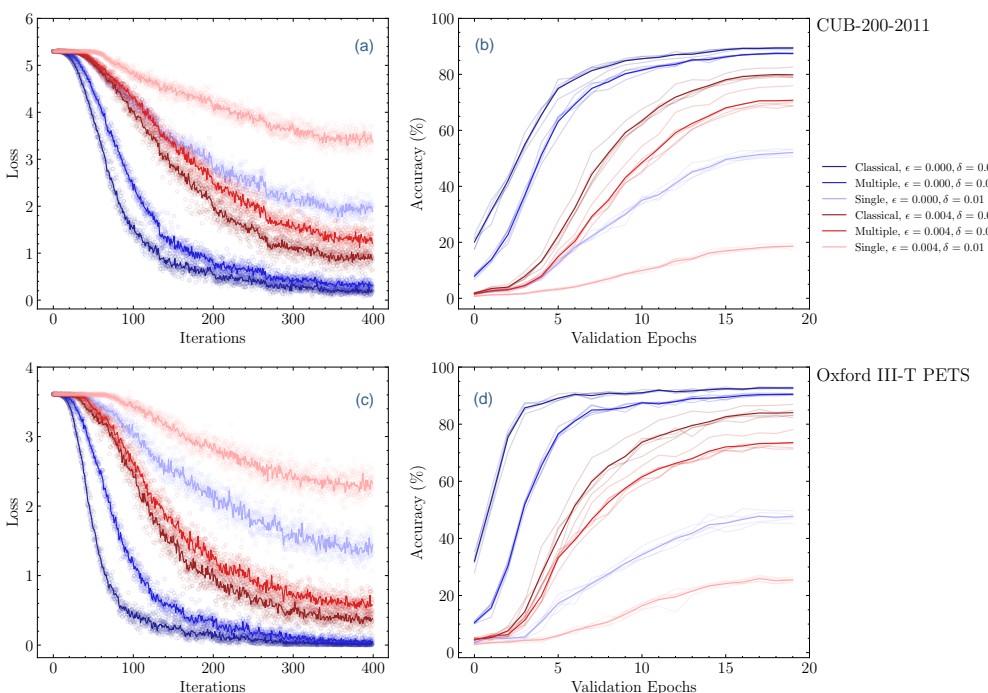

Figure 4: **Fine-tuning curve for the QViT with multiple/single-reuse strategies.** The loss and top 1 accuracy as the functions of steps of the QViT with the multiple/single-reuse strategies are shown in the left/right subfigures. Different colors represent different tomography strategies, tomography error $\delta$ or computing error $\epsilon$. Each experiment is repeated five times, and the data points or curves with higher transparency in the figure represent the results of a single experiment.

The fine-tuning convergence curves with different $\delta$ and $\epsilon$ for four datasets are shown in Figure 4 and Figure 12, showing both loss function and classification accuracy. Each numerical experiment is repeated 5 times to ensure reasonability. A more direct comparison between multiple and single-reuse strategies is given in Figure 5, which gives the classification accuracy under more parameter combinations. Corresponding data are listed in Table **??**. The two figures clearly show that the 'multiple' strategy always performs better, demonstrating enhanced convergence, tomography error resilience, and improved classification accuracy under every parameter combination. Another observation is that there is a threshold for both $\epsilon$ and $\delta$, around which the parameter fluctuation significantly influences the model performance while only delaying the convergence anywhere else. The 'multiple' strategy postpones the threshold of sampling so that the model can work with weaker conditions. As computing and tomography error grows, this strategy improves the worst performance, avoiding complete failure like the 'single' strategy does.

## 5 CONCLUSION

In this paper, we have proposed a novel strategy, measurement information multiple-reuse (MIMR), and an efficient multi-layer quantum vision transformer (QViT) model based on MIMR. While our QViT achieves exponential speedup for both forward pass and backpropagation processes, MIMR

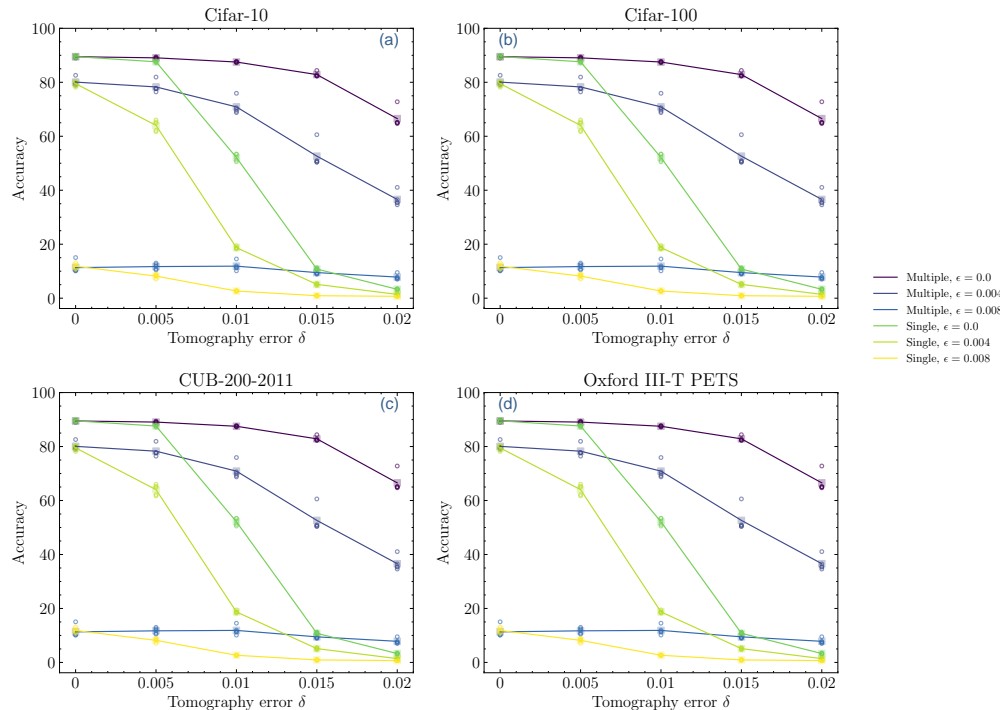

Figure 5: **The relationship curve between model performance, sampling, and computational error.** The classification accuracy after fine-tuning is significantly influenced by tomography error $\delta$ and computing error $\epsilon$. The relation is plotted in four figures, one for each dataset. Different colors consistently correspond to the two different measurement strategies and computing errors. The square markers represent the average of multiple numerical results, while individual experimental results are shown as hollow circles.

effectively addresses the critical bottleneck of information loss observed in previously proposed quantum deep neural network models, maximizing the utilization of measured information across layers. Benefiting from these advancements, we successfully constructed a transformer with more than 86 million parameters and numerically assessed its performance on real-world datasets for image classification. Our model demonstrated superior performance across four datasets—CUB-200-2011, CIFAR-10, CIFAR-100, and Oxford-IIIT PETS—achieving an average halving of the requirements for tomography precision, which implies a decrease in sampling costs to 25%. As a future direction, MIMR could be explored in other architectures to further demonstrate its generality and effectiveness as a universal strategy, independent of the specific QViT implementation. This study paves the way for future research toward exploring more efficient quantum deep neural networks, potentially leading to more scalable and powerful quantum artificial intelligence capable of tackling complex, real-world problems with unprecedented efficiency.

# 6 LIMITATIONS

Finally, we address some limitations of our work. First, the QViT requires fault-tolerant quantum computers and cannot run on NISQ devices. Second, the implementation of the QViT relies on qRAM, for which no effective physical realization currently exists. In Appendix C.5.2, we examine the practicality of qRAM in the context of the QViT. Although fault-tolerant quantum computers may become available in the next few decades, the physical realization of qRAM could be even more challenging. This suggests that the QViT may not be implementable on real quantum hardware for a considerable time.

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

## A  SYMBOLS, ABBREVIATIONS, AND DEFINITIONS

### A.1  MATHEMATICAL SYMBOLS

The mathematical symbols of this paper is shown in Table 3.

Table 3: Mathematical symbols

| Notation | Nomenclature |
|---|---|
| $X, X^{in}, X^{out}$ | Input/output data in each layer of QViT encoder. |
| $(d, n)$ | $d$:the dimension of each patch; $n$: the patch number |
| $x_i$ | The $i$-th column of $X$. |
| $P$ | Position embedding parameters |
| $h$ | Head number of the multi-head attention. |
| $L$ | QViT encoder layer depth. |
| $C$ | Cost function of the QViT. |
| $F$ | Parameters of a specific QViT layer. |
| $\|\cdot\|_F$ | Frobenius norm. |
| $\|\cdot\|_\infty$ | Infinite norm of a vector. |
| $\delta$ | Tomography error of the $l_\infty$ tomography. |

### A.2  ABBREVIATIONS

The abbreviations in this paper is shown in Table 4.

## B  THE PERFORMANCE COMPARISON OF THE MULTIPLE-REUSE AND THE SINGLE-REUSE STRATEGIES

In this section, we test the performance comparison of the multiple-reuse and the single-reuse strategies. In detail, we use randomly distributed vectors $x$ and $y$, defining $z = x + y$, where $\tilde{y}$ and

Table 4: Abbreviations

| Notation | Nomenclature |
|---|---|
| ViT | Vision Transformer |
| QViT | Quantum Vision Transformer |
| QRAM | Quantum Random Access Memory |
| QPos | Quantum Position Embedding |
| QHead | Quantum Head |
| QNorm | Quantum Norm |
| QAttn | Quantum Multi-head Attention |
| QAdd | Quantum Add |
| QFFN | Quantum Feedforward |
| QSave | Quantum Tomography |
| QLoad | Quantum Digital-Analog conversion |

$\tilde{z}$ represent the results of applying measurements to $y$ and $z$, respectively. $z_1 = \tilde{z}$ and $z_2 = \tilde{y} + x$ represent the outputs of the single-reuse and multiple-reuse strategies, respectively. We compare the cosine similarity between $z, z_1$, and $z, z_2$ and found that the latter is obviously better than the former, thus illustrating the effectiveness of the multiple-reuse strategy.

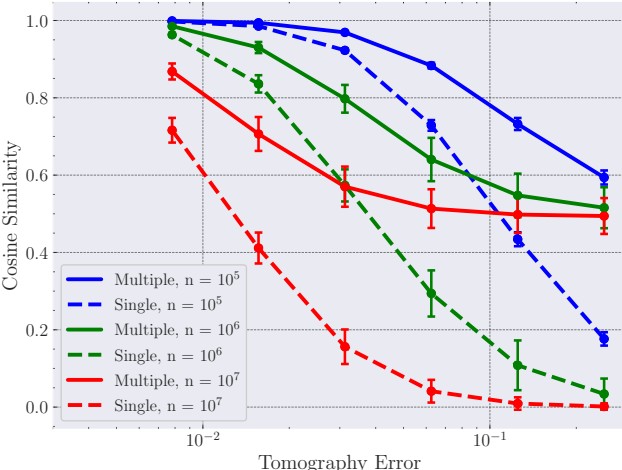

Figure 6: **The performance comparison of the multiple-reuse and the single-reuse strategies.** The target state is randomly generated, and the dimensions of states are $10^5$, $10^6$, and $10^7$, respectively. The results indicate that the multiple-reuse strategy achieves higher cosine similarity than single-reuse, particularly in scenarios with relatively large tomography errors.

## C BASICS OF QUANTUM COMPUTING

### C.1 QUANTUM ARITHMETIC

Quantum arithmetic is a fundamental module in quantum computing, involving the implementation of classical arithmetic operations using quantum circuits. The complexity of a specific quantum arithmetic operation is equivalent to that of the corresponding classical arithmetic operation, as detailed in Lemma C.1. Notably, the input of quantum arithmetic can be a superposition state, enabling the realization of the process:

$$\sum_{i=0}^{n-1} |x_i\rangle|0\rangle \to \sum_{i=0}^{n-1} |x_i\rangle|f(x_i)\rangle$$

with a complexity of $O(\text{polylog}(1/\epsilon))$.

**Lemma C.1** *Given a basic function $f(x) : \mathbb{R} \to \mathbb{R}$, there exists a quantum algorithm to implement quantum arithmetic $|x\rangle|0\rangle \to |x\rangle|\tilde{f}(x)\rangle$, where $|\tilde{f}(x) - f(x)| \le \epsilon$ and $\epsilon$ represents the computing accuracy. The gate complexity of the algorithm is $O(\text{polylog}(1/\epsilon))$.*

**Proof** *When the computing accuracy is $\epsilon$, the number of bits required is $O(\log(1/\epsilon))$, and the complexity of the corresponding classical arithmetic is $O(\text{polylog}(1/\epsilon))$. Classical arithmetic is constructed using general logic operations, which can be realized by basic quantum gates. Therefore, the target arithmetic can be implemented using basic quantum gates, with a gate complexity of $O(\text{polylog}(1/\epsilon))$.*

Next, we introduce existing quantum arithmetic algorithms required by our QViT except for the most basic quantum adders and multipliers.

### C.1.1 RECIPROCAL

We use the Newton method to calculate the reciprocal on a quantum computer Bhaskar et al. (2015). This target is expressed as:

$$|x, 0\rangle \to |x, \frac{1}{x}\rangle. \tag{3}$$

This can be approximately achieved through the following iteration:

$$x_{(k+1)} = x_{(k)}(2 - xx_{(k)}). \tag{4}$$

### C.1.2 ARC COSINE

The Quantum Function-value Binary Expansion method is chosen to calculate $\arccos$ Wang et al. (2020), which realizes approximately the transformation

$$|x, 0\rangle \to |x, \arccos x\rangle. \tag{5}$$

The iteration reads

$$x_{(0)} = x, \quad x_{(k+1)} = \begin{cases} 2x_{(k)}^2 - 1, & x_{(k)} > 0, \\ 1 - 2x_{(k)}^2, & x_{(k)} \le 0. \end{cases} \tag{6}$$

### C.1.3 RELU

The ReLU function,

$$f(x) = \max(0, x), \tag{7}$$

can be implemented directly by a controlled quantum adder. We represent the signed number $x$ as a bit string $x_0|x|$, where $x_0$ is the sign bit, and $|x|$ is the magnitude in either $s$-qubit true form or two's complement. Apply a quantum adder and CNOT controlled by $x_0$ and we have

$$|x_0, |x|\rangle \otimes |0^{\otimes s}\rangle \to |x_0, |x|\rangle \otimes |0, x_0 \times |x|\rangle \tag{8}$$
$$\to |x_0, |x|\rangle \otimes |x_0, x_0 \times |x|\rangle \tag{9}$$
$$= |x_0, |x|\rangle \otimes |\max(0, x)\rangle. \tag{10}$$

### C.2 QUANTUM TOMOGRAPHY

**Theorem C.2** *[$l_\infty$ vector state tomography Kerenidis et al. (2020b)]Given access to unitary $U$ such that $U|0\rangle = |x\rangle$ and its controlled version in time $T(U)$, there is a tomography algorithm with time complexity $O(T(U)\frac{\log d}{\delta^2})$ that produces unit vector $\tilde{X} \in \mathbb{R}^d$ such that $\|\tilde{X} - x\|_\infty \le \delta$ with probability at least $(1 - 1/\text{poly}(d))$.*

## C.3 QUANTUM DIGITAL-ANALOG CONVERSION

In the QViT implementation process, we utilize quantum digital-analog conversion (QDAC), as introduced in Mitarai et al. (2019). We present the main results of QDAC in Lemma C.3. It's worth noting that the expression provided in Lemma C.3 may not align completely with the one in Mitarai et al. (2019). Therefore, we provide the proof of Lemma C.3 to clarify any discrepancies.

**Lemma C.3** *(Generalized QDAC) Given the D-Encoding of $x \in \mathbb{R}^n$, let $f_x = [f(x_1), f(x_2), \cdots, f(x_n)]$, where $f(x_i)$ represents some basic functions of $x_i$. Then, there exists an algorithm to prepare the A-Encoding of $f_x$ with $\Omega(1)$ success probability. The query complexity to the D-Encoding of $x$ is $O(1/\sqrt{\nu + \mu^2})$, where $\nu$ and $\mu$ are the variance and mean value of $f_x/\|f_x\|_\infty$, respectively.*

**Proof** *The preparation process of $|f_x\rangle$ is as follows:*

*(1) Prepare superposition state $\frac{1}{\sqrt{n}} \sum_{i=1}^n |i\rangle$.*

*(2) Execute $U$ to obtain $\frac{1}{\sqrt{n}} \sum_{i=1}^n |i\rangle |x_i\rangle$.*

*(3) Add an ancilla qubit and perform rotation operations controlled by $|x_i\rangle$, resulting in the quantum state:*

$$\frac{1}{\sqrt{n}} \sum_{i=1}^n |i\rangle |x_i\rangle (\frac{f(x_i)}{C}|0\rangle + \sqrt{1 - \frac{f^2(x_i)}{C^2}}|1\rangle). \tag{11}$$

*(4) Measure the ancilla qubit to $|0\rangle$ and uncompute $|x_i\rangle$, yielding:*

$$\frac{1}{\|f_x\|} \sum_{i=1}^n f(x_i)|i\rangle. \tag{12}$$

*The success probability of this process is $p = \frac{\sum_{i=1}^n f^2(x_i)}{nC^2} = \nu + \mu^2$, where $\nu$ and $\mu$ are the variance and mean value of $f_x/\|f_x\|_\infty$. Utilizing the amplitude amplification algorithm, $|f_x\rangle$ can be prepared by querying $U$ $O(1/\sqrt{\nu + \mu^2})$ times.*

In QDAC, $1/\sqrt{\nu + \mu^2}$ is related to specific data distribution. Different specific problems correspond to different $1/\sqrt{\nu + \mu^2}$. Therefore, in subsequent analysis, we ignore the influence of $1/\sqrt{\nu + \mu^2}$ and treat it as a constant.

## C.4 BLOCK-ENCODING

Block-encoding offers a methodology for executing non-unitary operations in the domain of quantum computing Gilyén et al. (2019); Martyn et al. (2021). This technique involves encapsulating a non-unitary operator $A$ within a unitary matrix $U_A$, a process referred to as the block-encoding of $A$. The operator $A$ can then be applied probabilistically through the execution of its block-encoded counterpart $U_A$.

**Definition C.4** *(Block-encoding) Suppose that $A$ is an $s$-qubit operator, $\alpha, \epsilon \in \mathbb{R}_+$ and $a \in \mathbb{N}$, then we say that the $(s + a)$-qubit unitary $U$ is an $(\alpha, a, \epsilon)$-block-encoding of $A$, if*

$$\|A - \alpha(\langle 0|^{\otimes a} \otimes I)U(|0\rangle^{\otimes a} \otimes I)\| \le \epsilon. \tag{13}$$

In our work, we construct the block-encoding of $X$ by querying the D-Encoding of $X$. The result is presented in Lemma C.5.

**Lemma C.5** *Given D-Encoding of $X = [x_0, x_1, \cdots, x_{n-1}] \in \mathbb{R}^{d \times n}$, a $(\|X\|_F, \lceil \log(d + n) \rceil, \epsilon)$-block-encoding of $X$ can be built by querying qRAM $\tilde{O}(d)$ times.*

**Proof** *First, by querying the D-Encoding $d$ times, we construct the following unitary transformations:*

$$U_R : |0\rangle |j\rangle \mapsto |x_j\rangle |j\rangle, \tag{14}$$

$$V : |0\rangle |j\rangle \mapsto |y_j\rangle |j\rangle, y_j = \|x_j\|. \tag{15}$$

*Then, we utilize QDAC to build:*

$$U_L : |i\rangle|0\rangle \mapsto |i\rangle\frac{\sum_{j=1}^n \|x_j\|_F|j\rangle}{\|X\|_F}, \tag{16}$$

*We have*

$$|\psi_i\rangle = U_R|i\rangle|0\rangle, |\phi_j\rangle = U_L|0\rangle|j\rangle, \langle\phi_j|\psi_i\rangle = \frac{X_{ij}}{\|X\|_F}. \tag{17}$$

*Therefore, $U_L^\dagger U_R$ is a $(\|X\|_F, \lceil\log(d+n)\rceil, \epsilon)$-block-encoding of $X$, the query complexity to the qRAM is $\tilde{O}(d)$.*

## C.5 Quantum Random Access Memory

### C.5.1 Introduction

In this section, we introduce quantum random access memory (qRAM) Giovannetti et al. (2008), a quantum architecture fundamental to our framework. QRAM serves as a generalization of classical RAM, leveraging quantum mechanical properties to enhance computational efficiency.

In classical RAM, a discrete address $i$ is provided as input, retrieving the memory element $x_i$ stored at that location. Conversely, in qRAM, a quantum superposition of different addresses $|\psi_{\text{in}}\rangle$ is input, and qRAM returns an entangled state $|\psi_{\text{out}}\rangle$ where each address is correlated with the corresponding memory element:

$$|\psi_{\text{in}}\rangle = \sum_{i=0}^{N-1} \alpha_i|i\rangle_A|0\rangle_D \xrightarrow{\text{qRAM}} |\psi_{\text{out}}\rangle = \sum_{i=0}^{N-1} \alpha_i|i\rangle_A|x_i\rangle_D, \tag{18}$$

where $N$ is the size of the data vector $x$, and the superscripts $A$ and $D$ denote "address" and "data" respectively.

While we have characterized our QViT as the quantum deep learning framework in the fault-tolerant era, it is still imperative to incorporate the simulation of noisy qRAM. It is frequently used in QSave and QLoad in QViT. Within this framework, the primary role of qRAM is to retrieve pixel information stored in a massive matrix of size $2^{20}$ by $2^{10}$. Each pixel can hold either 32 or 64 bits, necessitating a $(30, 64)$ or $(30, 32)$-qRAM configuration. Our numerical simulations demonstrate promising results. For a $(30, 64)$-qRAM configuration, we observe an average state fidelity of $87\%$. This fidelity increases to $91\%$ for the $(30, 32)$-qRAM configuration.

### C.5.2 Practicality of qRAM used in QViT

The practicality of qRAM has been investigated on such a scale under our numerical experiments. Prior research indicates that qRAM infidelity scales as $O(n(n+k))$, where $n$ represents the address size and $k$ denotes the word length. This implies that infidelity exhibits quadratic growth with respect to address size for a fixed $k$ and increases with word length for a fixed address size $n$. Based on these established relationships, our experiments employed data with a fixed word length $k$ to maintain consistency with the established infidelity relation. Subsequently, we extrapolated the findings to the case of $n = 30$. Using these extrapolated data $F(30, k)$, we employed a linear function to predict the infidelity value when $k = 64$.

All simulations were conducted under a controlled environment with $10^{-5}$ damping noise. We successfully simulated qRAM configurations ranging from $(20, 20)$ and below. The observed infidelities agree to the $O(n(n+k))$ relationship, demonstrating a quadratic dependence on address size $n$ for a fixed word length $k$, as shown in Figure 7 and Figure 8.

We leveraged the data obtained from these simulations to derive polynomial expressions that accurately capture the relationship between infidelity and address size. Subsequently, these quadratic expressions were utilized to extrapolate and predict the infidelities of $(30, k)$-QRAM for a range of $k$ values from 1 to 20. As a result, we have obtained a comprehensive set of predicted QRAM infidelities for $(30, k)$ configurations, where $k$ ranges from 1 to 64, as shown in Figure 9.

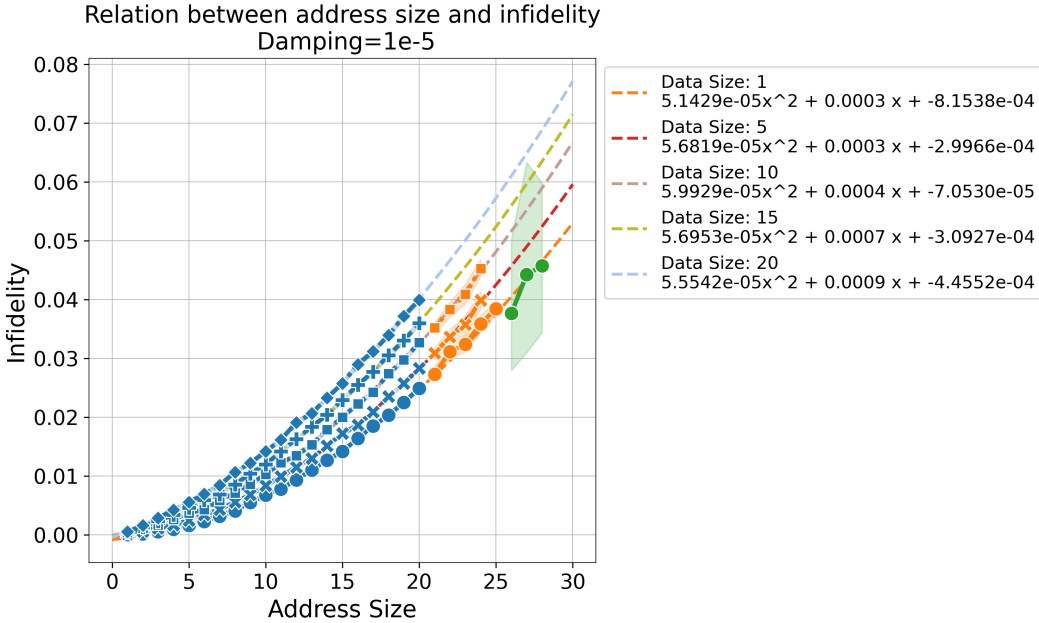

Figure 7: The figure shows the relation between address size and infidelity with the fixed k. The blue points are from the numerical experiments with 100 repetitions and each repetition of experiments consists of 1000 shots. The orange points are experiments with 10 repetitions with 1000 shots each. The Green ones are experiments with 10 repetitions with 10 shots each. The fitting function is completely decided by the data of blue points.

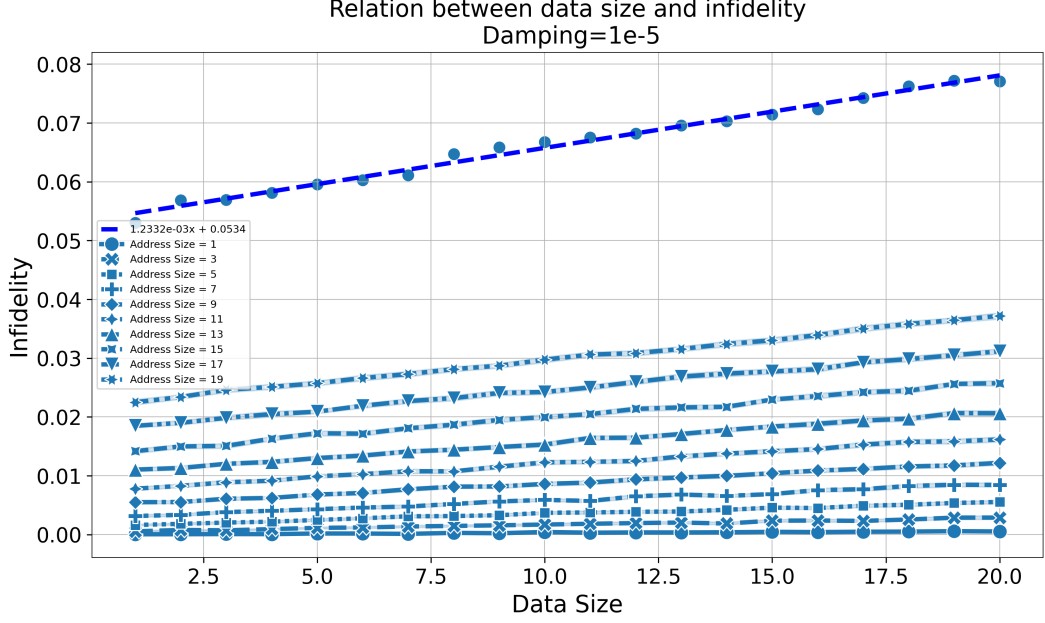

Figure 8: The figure presents the scattering points and sketches the linear relations between data size $k$ and the infidelity.

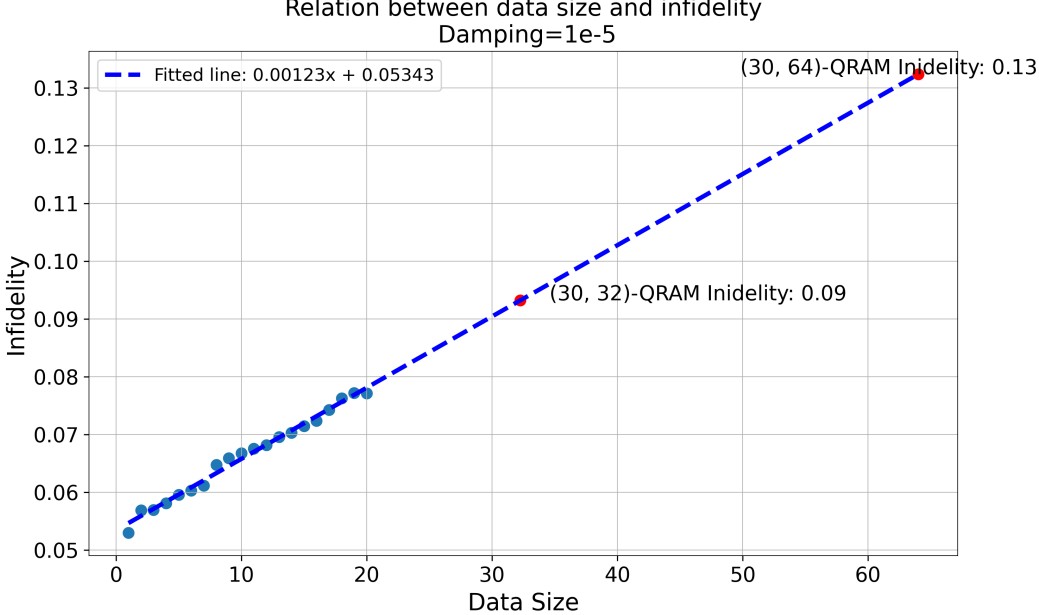

Figure 9: The figure presents the fitting function and annotates the predicted fidelities of $(30, 32)$-QRAM and $(30, 64)$-QRAM.

## D    IMPLEMENTATION DETAILS OF THE QVIT

In this section, we introduce details of the implementation of the QViT. Appendix D.1 gives an overview, and concrete implementations of the forward pass and backpropagation of each layer are included in the rest of this section.

### D.1    OVERVIEW

Some layers within the QViT utilize D-Encoding for their input/output, necessitating the construction of corresponding D-Encoding operations. In the QViT, the data is stored in the QRAM; for a given $X \in \mathbb{R}^{d \times n}$ stored in QRAM, the D-Encoding of $X$ is realized through QRAM querying.

Then, we present the process of implementing the QViT's forward pass and backpropagation, as outlined in Algorithms 1 and 2. As defined in Eq. (2), building the D-Encoding of $X \in R^{n \times d}$ means building $O_D(X)$ which satisfies

$$O_D(X)|i,j\rangle|0\rangle = |i,j\rangle|X_{i,j}\rangle, i = 0, 1, \cdots, n-1, j = 0, 1, \cdots, d-1. \tag{19}$$

In Algorithms 1 and 2, the D-Encoding is built by quantum arithmetic. Because QPos, QNorm, QAdd, and QFFN layer can be viewed as $n$ $d$-dimensional operations, the D-Encoding in these layers can be built by $O(d)$ basic quantum arithmetic operations.

The QAttn layer is different from other layers. The input of the QAttn layer is the D-Encoding of $X^{in}$, we first prepare the amplitude encoding state $|X^{out}\rangle$. Then, we sample $|X^{out}\rangle$ with $l_\infty$ tomography and construct the D-Encoding of $X^{out}$ with the sampled results.

Then, we establish the D-Encoding for the input to the next QAttn layer using quantum arithmetic. The corresponding quantum circuit is depicted in Figure 10. By querying $X^0, \tilde{Y}^0$, through to $\tilde{Y}^l$ twice, we construct the D-Encoding of $X^{l+1}$, which serves as the input for the $l + 1$-th QAttn layer. As shown in Figure 10, the initial input and all intermediate measurement data are propagated forward, reflecting the MIMR scheme and thereby enhancing the utilization of intermediate information.

We further explicate the implementation of each layer in the QViT.

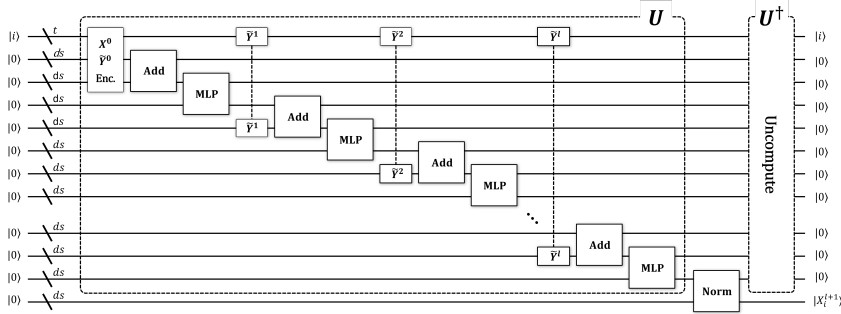

Figure 10: **Process for constructing D-Encoding of** $X^{l+1}$**.** $s$ represents the precision of quantum arithmetic, it can be 8, 16, etc. $i$ denotes the column number. $X^0$ is the input for the initial quantum attention layer, while $\tilde{Y}^j$ denotes the tomography results from the output of the $j$-th QAttn layer, where $j$ ranges from 0 to $l$. Blocks labeled with "$X^0$" or "$\tilde{Y}^j$" indicate queries to the QRAM storing the respective "$X^0$" or "$\tilde{Y}^j$". Other blocks represent the $d$-dimensional quantum arithmetic operations, including "Add", "MLP", and "Norm". Specifically, "MLP" involves "Norm", "FFN", and "Add". The block labeled "$U^\dagger$" denotes the uncomputing block.

---

**Algorithm 1** Forward pass of QViT.

---

1: **Input:** data $X$.
2: **Output:** classification label of $X$.
3: QPos: Build D-Encoding of $X^{out} = X^{in} + P$, where $P$ represents position embedding.
4: **for** $i = 0, 1, 2, \cdots, L - 1$ **do**
5:     QNorm: Build D-Encoding of $X^{out} = \text{Norm}(X^{in})$.
6:     QAttn: Prepare the A-Encoding state $|X^{out}\rangle$ where $X^{out}$ is the output of the multi-head attention. Then, sample $|X^{out}\rangle$ with $l_\infty$ tomography and construct the D-Encoding of $X^{out}$ with the sampled results.
7:     QAdd: Build the D-Encoding of $X^{out} = X^{(1)} + X^{(2)}$, where $X^{(1)}$ represents the output of step **6**, and $X^{(2)}$ represents the input of step **5**.
8:     QNorm: Build the D-Encoding of $X^{out} = \text{Norm}(X^{in})$.
9:     QFFN: Build the D-Encoding of $X^{out} = W_2 f(W_1 X^{in} + b_1) + b_2$.
10:    QAdd: Build the D-Encoding of $X^{out} = X^{(1)} + X^{(2)}$, where $X^{(1)}$ represents the output of step **9**, and $X^{(2)}$ represents the input of step **8**.
11: QHead: Prepare A-Encoding state $|X^{out}\rangle$ where $X^{out} = W x_0^{in} + b$, then sample $|X^{out}\rangle$ and obtain the classification label from the sampled results.

---

### D.2   POSITION EMBEDDING

The formulation for Position Embedding is given by $X^{out} = X^{in} + P$, where $P \in \mathbb{R}^{d \times n}$ represents the position embedding parameters.

#### D.2.1   FORWARD

The input and output are the D-Encoding of $X^{in}$ and $X^{out}$, respectively. The D-Encoding of $P$ is built through a single query to the QRAM. Therefore, the D-Encoding of $X^{out}$ is constructed by querying the D-Encoding of $X^{in}$ and $P$ once.

#### D.2.2   BACKPROPAGATION

The input is the D-Encoding of $\frac{\partial C}{\partial X^{out}}$, and the output is the sampled $\frac{\partial C}{\partial P}$. We have

$$\frac{\partial C}{\partial P} = \frac{\partial C}{\partial X^{out}}, \tag{20}$$

therefore, we obtain the D-Encoding of $\frac{\partial C}{\partial P}$. Subsequently, we employ QDAC to prepare the A-Encoding state $|\frac{\partial C}{\partial P}\rangle$ and obtain the sampled $\frac{\partial C}{\partial P}$ through $l_\infty$ tomography.

---

**Algorithm 2** Backpropagation of QViT.

---

1: **Input:** data $X$, forward pass results.
2: **Output:** Sampled $\frac{\partial C}{\partial F}$, where $F$ represents parameters in the QViT.
3: Build D-encoding of $\frac{\partial C}{\partial X^{out}}$ through the forward pass results, where $X^{out}$ is the output of the QHead.
4: QHead: (1) Prepare A-Encoding state $|\frac{\partial C}{\partial F}\rangle$, where $F$ represents parameters of the QHead, then obtain the sampled $\frac{\partial C}{\partial F}$. (2) Build D-Encoding of $\frac{\partial C}{\partial X^{in}}$.
5: **for** $i = L - 1, L - 2, \cdots, 1, 0$ **do**
6:     QAdd: Build D-encoding of $\frac{\partial C}{\partial X^{(1)}}, \frac{\partial C}{\partial X^{(2)}}$.
7:     QFFN: (1) Prepare A-Encoding state $|\frac{\partial C}{\partial F}\rangle$, where $F$ represents the parameters of the QFFN, then obtain the sampled $\frac{\partial C}{\partial F}$. (2) Build D-Encoding of $\frac{\partial C}{\partial X^{in}}$.
8:     QNorm: Build D-Encoding of $\frac{\partial C}{\partial X^{in}}$.
9:     QAdd: Build D-encoding of $\frac{\partial C}{\partial X^{(1)}}, \frac{\partial C}{\partial X^{(2)}}$.
10:     QAttn: Prepare A-Encoding states $|\frac{\partial C}{\partial X^{in}}\rangle$ and $|\frac{\partial C}{\partial F}\rangle$, $F$ represents parameters of the QAttn, then obtain sampled $\frac{\partial C}{\partial F}$ and $\frac{\partial C}{\partial X^{in}}$. Next, build D-Encoding of $\frac{\partial C}{\partial X^{in}}$.
11:     QNorm: Build D-Encoding of $\frac{\partial C}{\partial X^{in}}$.
12: QPos: Prepare A-Encoding state $\frac{\partial C}{\partial P}$ and obtain the sampled $\frac{\partial C}{\partial P}$.

---

### D.3 QNORM LAYER

The norm layer is formulated as $X^{out} = \text{Norm}(X^{in})$, detailed by:

$$X^{out} = [\frac{x_1^{in} - \mu_1}{\sigma_1}, \frac{x_2^{in} - \mu_2}{\sigma_2}, \cdots, \frac{x_n^{in} - \mu_n}{\sigma_n}], \tag{21}$$

where $\mu_i = \frac{\sum_{j=1}^{d} x_{ij}^{in}}{d}, \sigma_i^2 = \frac{\sum_{j=1}^{d} (x_{ij}^{in} - \mu_i)^2}{d}$.

#### D.3.1 FORWARD

In the QNorm layer, the D-Encoding of $X^{in}$ serves as the input, producing the D-Encoding of $X^{out}$ as output. For each $x_i^{in} \in \mathbb{R}^d$ with $i = 0, 1, \cdots, n-1$, both $\mu_i$ and $\sigma_i$ can be computed by querying the D-Encoding of $X^{in}$ $d$ times, which means the following two operations:

$$|i\rangle|0\rangle \mapsto |i\rangle|\mu_i\rangle, |i\rangle|0\rangle \mapsto |i\rangle|\sigma_i\rangle. \tag{22}$$

Following this, the D-Encoding of $X^{out}$ is constructed by querying the operations defined in Eq. (22) and the D-Encoding of $X^{in}$.

#### D.3.2 BACKPROPAGATION

During the backpropagation procedure, we can establish the relationship between the D-Encoding of $\frac{\partial C}{\partial X^{in}}$ and the D-Encoding of $\frac{\partial C}{\partial X^{out}}$. This relationship is formulated as follows:

$$\frac{\partial C}{\partial x_i^{in}} = \frac{\partial C}{\partial x_i^{out}} \frac{\partial x_i^{out}}{\partial x_i^{in}}, \ \frac{\partial x_i^{out}}{\partial x_i^{in}} = \frac{dI - \vec{1}}{d\sigma_i} - \frac{(x_i^{in} - \mu_i)(x_i^{in} - \mu_i)^T}{d\sigma_i^3}, \tag{23}$$

where $\vec{1}$ represents a matrix in which all elements equal to $1$. By applying the above equation, the D-Encoding of $\frac{\partial C}{\partial x_i^{in}}$ is obtained by querying the D-Encoding of both $\frac{\partial C}{\partial X^{out}}$ and $X^{in}$ $d$ times.

### D.4 QUANTUM ATTENTION

The attention operation is defined as:

$$\text{Attention}(X^{in}, W_q, W_k, W_v) = VA', A' = \text{softmax}(\frac{A}{\sqrt{d}}), A = K^T Q, [Q, K, V] = [W_q, W_k, W_v]X^{in}, \tag{24}$$

where $W_q, W_k, W_v \in \mathbb{R}^{d \times d}$, and the softmax function is applied column-wise.

The multi-head attention is defined as:

$$X^{out} = W\text{Concat}(H_0, H_1, \cdots, H_{h-1}), H_m = \text{Attention}(X, W_{qm}, W_{km}, W_{vm}), \quad (25)$$

where $W = [W_0, W_1, \cdots, W_{h-1}] \in \mathbb{R}^{d \times hd}$; $W_{qm}, W_{km}, W_{vm} \in R^{d \times d}$, for $m = 0, 1, \cdots, h - 1$.

### D.4.1 FORWARD

In the quantum attention layer, the process begins with the D-Encoding of $X^{in}$. The aim is to prepare the A-Encoding state $|X^{out}\rangle$, followed by sampling $|X^{out}\rangle$ using $l_\infty$ tomography, and finally build the D-Encoding of $X^{out}$ by querying the tomography results.

First, we prepare the A-Encoding of $A^{'}$ with Lemma D.1. Then we build the D-Encoding of $V$ by querying the D-Encoding of $X$ $d$ times and $(\|V\|_F, \lceil \log(d + n) \rceil, \epsilon)$-block-encoding of $V$ following the method described in Lemma C.5. Finally, we apply the block-encoding of $V$ on $|A^{'}\rangle$ and measure the ancilla qubits to $|0\rangle$, resulting in:

$$|X^{out}\rangle = \frac{1}{\|X^{out}\|} \sum_j V|A^{'}_{:,j}\rangle|j\rangle, \quad (26)$$

where $A^{'}_{:,j}$ represents the $j$-th column of $A^{'}$.

Furthermore, multi-head attention is constructed for $l = 0, 1, \cdots, h - 1$, executing quantum attention in parallel to achieve:

$$|H\rangle = \frac{1}{\|H\|} \sum_{l=0}^{h-1} |l\rangle \otimes \|H_l\| |H_l\rangle, \quad (27)$$

where $H_l = \text{Attention}(Q^i, K^i, V^i)$, $V^i = W_{vi}X$, $K^i = W_{ki}X$, $Q^i = W_{qi}X$, and $H = \text{Concat}(H_0, H_1, \cdots, H_{h-1})$. Then we construct realize $W$ operation with block-encoding technique and obtain

$$|X^{out}\rangle = W|H\rangle. \quad (28)$$

Ultimately, $X^{out}$ is sampled $\widetilde{O}(\frac{\log(dn)}{\delta^2})$ times, with the D-Encoding of $X^{out}$ being constructed from querying the tomography results.

**Lemma D.1** *Given D-Encoding of $X \in \mathbb{R}^{d \times n}$, $W_q, W_k \in \mathbb{R}^{d \times d}$, $A = X^T W_k^T W_q X$, $A^{'} = softmax(\frac{A}{\sqrt{d}})$, then there exists a quantum algorithm to prepare $|A^{'}\rangle = \frac{1}{\|A^{'}\|} \sum_{i,j} A^{'}_{ij} |i\rangle|j\rangle$ with $\Omega(1)$ success probability. The query complexity to the D-Encoding of $X$ is $\tilde{O}(d/\epsilon)$.*

**Proof** *Firstly, the element of $A$ is calculated as $A_{ij} = x_i^T W_k^T W_q x_j$. Therefore, the D-Encoding of $A$ is built by querying the D-Encoding of $X$ $2d$ times. Then we define matrix $A^{''}$ which satisfies $A^{''}_{ij} = e^{A_{ij}/\sqrt{d}}$ and prepare state*

$$|\sqrt{A^{''}}\rangle = \frac{1}{\|\sqrt{A^{''}}\|} \sum_{i,j} \sqrt{A^{''}_{ij}} |i\rangle|j\rangle \quad (29)$$

*with QDAC. For a specific $j^{'}$, the state $|\sqrt{A^{''}}\rangle$ manifests as*

$$|\sqrt{A^{''}}\rangle = \frac{\sqrt{b_{j'}}}{\|\sqrt{A^{''}}\|} \left( \frac{1}{\sqrt{b_{j'}}} \sum_i \sqrt{A^{''}_{ij'}} \right) |j^{'}\rangle + |\psi^\perp\rangle, \quad (30)$$

*where $\langle j^{'}|\psi^\perp\rangle = 0$. Amplitude estimation algorithm Brassard et al. (2002) is then employed to determine $\frac{\sqrt{b_{j'}}}{\|\sqrt{A^{''}}\|}$. Since $\|\sqrt{A^{''}}\|$ is known from $|\sqrt{A^{''}}\rangle$'s preparation, $b_{j'}$ is obtained. By executing amplitude estimation in parallel for each $j^{'}$, we realize the following operation*

$$|\sqrt{A^{''}}\rangle|0\rangle \rightarrow \frac{1}{\|\sqrt{A^{''}}\|} \sum_{i,j} \sqrt{A^{''}_{ij}} |i\rangle|j\rangle|b_j\rangle \rightarrow |\psi\rangle = \frac{1}{\|\sqrt{A^{''}}\|} \sum_{i,j} \sqrt{A^{''}_{ij}} |i\rangle|j\rangle|\sqrt{A^{''}_{ij}/b_j}\rangle, \quad (31)$$

*with a query complexity to $|\sqrt{A''}\rangle$ of $O(1/\epsilon)$. Finally, we use QDAC to prepare*

$$\frac{1}{\|A'\|}\sum_{i,j}\sqrt{A''_{ij}}/b_j\sqrt{A''_{ij}}|i\rangle|j\rangle = |A'\rangle. \tag{32}$$

*In summary, the query complexity to the D-Encoding of $X$ is $\tilde{O}(d/\epsilon)$.*

**Lemma D.2** *(Forward pass of QAttn) Given the D-Encoding of $X$, $W_q, W_k, W_v \in \mathbb{R}^{h\times d\times d}$, $W \in \mathbb{R}^{d\times hd}$, where $h$ denotes the head number, then there exists a quantum algorithm to implement the QAttn layer. This process constructs the D-Encoding of the layer output, with the query complexity to the D-Encoding of $X$ being $\tilde{O}(\frac{\log(n)hd}{\epsilon\delta^2})$.*

**Proof** *First, for each $m = 0, 1, \cdots, h-1$, $|A'_m\rangle$ is prepared by querying the D-Encoding of $X$ $\tilde{O}(d/\epsilon)$ times. For each column $(V_m)_i$, which is computed using $x_i$, by lemma C.5, a $(\|V_m\|_F, \lceil\log(d+n)\rceil, \epsilon)$-block-encoding of $V_m$ is constructed by querying D-Encoding of $X$ $\tilde{O}(d/\sqrt{\nu + \mu^2})$ times, where $\nu$ and $\mu$ are variance and mean of $y/\|y\|_\infty$, respectively, with $y = [(V_m)_0, (V_m)_1, \cdots, (V_m)_{n-1}]$.*

*Then, $|H_m\rangle$ is prepared by querying both the preparation of $|A'_m\rangle$ and the block-encoding of $V_m$. Therefore, $|H_m\rangle$ is prepared by querying the D-Encoding of $X$ $\tilde{O}(d/\epsilon)$ times and cumulatively, $\tilde{O}(hd/\epsilon)$ for all heads in the construction of $|H\rangle$.*

*Ultimately, $|X^{out}\rangle$ is prepared by operating $W_F$ on $|H\rangle$. Finally, we sample $|X^{out}\rangle$ $\tilde{O}(\frac{\log(dn)}{\delta^2})$ times and build the corresponding D-Encoding with the tomography results. In summary, the query complexity to the D-Encoding of $X$ is $\tilde{O}(\frac{hd\log(n)}{\epsilon\delta^2})$.*

### D.4.2 BACKPROPAGATION

In the backpropagation process, the input is the D-Encoding of $\frac{\partial C}{\partial X^{out}}$. The procedure begins with preparing the A-Encoding state $|\frac{\partial C}{\partial X^{in}}\rangle$ and $|\frac{\partial C}{\partial F}\rangle$, where $F$ denotes the parameters of this layer. Subsequently, these two A-Encoding states are sampled with $l_\infty$ tomography, building the D-Encoding of $\frac{\partial C}{\partial X^{in}}$ based on the tomography results of $\frac{\partial C}{\partial X^{in}}$.

Firstly, we prepare the state $|\frac{\partial C}{\partial F}\rangle$, where $F$ contains $W$, $W_{vm}$, $W_{qm}$ and $W_{km}$ for $m = 0, 1, \cdots, h-1$. The derivative of $C$ with respect to $W$ is expressed as:

$$\frac{\partial C}{\partial W} = \frac{\partial C}{\partial X^{out}}\frac{\partial X^{out}}{\partial W}, (\frac{\partial X^{out}}{\partial W})_{ijkl} = \delta_{ik}H^T_{jl}, \tag{33}$$

For $m = 0, 1, \cdots, h-1$, the derivative with respect to $\frac{\partial C}{\partial W_{vm}}$ and $\frac{\partial C}{\partial W_{qm}}$ are given by:

$$\frac{\partial C}{\partial W_{vm}} = \frac{\partial C}{\partial X^{out}}\frac{\partial X^{out}}{\partial H_m}\frac{\partial H_m}{\partial V_m}\frac{\partial V_m}{\partial W_{vm}}, \tag{34}$$

$$(\frac{\partial X^{out}}{\partial H_m})_{ijkl} = \delta_{jl}(W_m)_{ik}, (\frac{\partial H_m}{\partial V_m})_{ijkl} = \delta_{ik}(A'_m)^T_{jl}, (\frac{\partial V_m}{\partial W_{vm}})_{ijkl} = \delta_{ik}X^{in}_{lj}, \tag{35}$$

$$\frac{\partial C}{\partial W_{qm}} = \frac{\partial C}{\partial X^{out}}\frac{\partial X^{out}}{\partial H_m}\frac{\partial H_m}{\partial A'_m}\frac{\partial A'_m}{\partial A_m}\frac{\partial A_m}{\partial Q_m}\frac{\partial Q_m}{\partial W_{qm}}, \tag{36}$$

$$(\frac{\partial H_m}{\partial A'_m})_{ijkl} = \delta_{jl}(V_m)_{ik}, (\frac{\partial A'_m}{\partial A_m})_{ijkl} = \frac{1}{\sqrt{d}}\delta_{jl}(\delta_{ik}(A'_m)_{ij} - (A'_m)_{ij}(A'_m)_{kj}), \tag{37}$$

$$(\frac{\partial A_m}{\partial Q_m})_{ijkl} = \delta_{jl}(K^T)_{ik}, (\frac{\partial Q_m}{\partial W_{qm}})_{ijkl} = \delta_{ik}X^{in}_{lj}. \tag{38}$$

The expression of $\frac{\partial C}{\partial W_{km}}$ is similar to $\frac{\partial C}{\partial W_{qm}}$ introduced in Eq. (36). From Eq. (33) to (38), each component of $\frac{\partial C}{\partial F}$ consists of $H$, $W$, $A'_m$, $Q_m$, $K_m$, $V_m$, or $X^{in}$. The corresponding D-Encoding, A-Encoding or block-encoding of these matrices are introduced before. Therefore, we can prepare A-Encoding of each component of $\frac{\partial C}{\partial F}$ with quantum linear algebra, that is, prepare A-Encoding of $\frac{\partial C}{\partial F}$. After sampling the A-Encoding state $|\frac{\partial C}{\partial F}\rangle$, parameters are updated based on the sampled results.

Next, we consider $\frac{\partial C}{\partial X^{in}}$, which is given by:

$$\frac{\partial C}{\partial X^{in}} = \frac{\partial C}{\partial X^{out}} \frac{\partial X^{out}}{\partial H} \frac{\partial H}{\partial X^{in}}. \tag{39}$$

For $m = 0, 1, \cdots, h-1$,

$$\frac{\partial H_m}{\partial X^{in}} = \frac{\partial V_m}{\partial X^{in}} A'_m + V_m \frac{\partial A'_m}{\partial X^{in}}, \tag{40}$$

$$\frac{\partial A'_m}{\partial X^{in}} = \frac{\partial A'_m}{\partial A_m}\left(\frac{\partial K_m^T}{\partial X^{in}} Q_m + K_m^T \frac{\partial Q_m}{\partial X^{in}}\right), \left[\frac{\partial V_m}{\partial X^{in}}, \frac{\partial Q_m}{\partial X^{in}}, \frac{\partial K_m}{\partial X^{in}}\right]_{ijkl} = \delta_{jl}[W_{vm}, W_{qm}, W_{km}]_{ik}. \tag{41}$$

Similar to $\frac{\partial C}{\partial F}$, $\frac{\partial C}{\partial X^{in}}$ also consists of $\frac{\partial C}{\partial X^{out}}$, $W$, $A'$, $Q_m$, $K_m$, $V_m$, $W_{vm}$, $W_{qm}$, and $W_{km}$, and the corresponding D-Encoding, A-Encoding or block-encoding of these matrices are introduced before. Therefore, we can prepare A-Encoding of $\frac{\partial C}{\partial X^{in}}$ and sample $|\frac{\partial C}{\partial X^{in}}\rangle$ with $l_\infty$ tomography, then we build D-Encoding of $\frac{\partial C}{\partial X^{in}}$ with the sampled results. The cost associated with backpropagation of QAttn can be summarized in the following lemma.

**Lemma D.3** *(Backpropagation of QAttn) Given D-Encoding of $X^{in}$, $X^{out}$ and $\frac{\partial C}{\partial X^{out}}$, $W_q, W_k, W_v \in \mathbb{R}^{h \times d \times d}$, $W \in \mathbb{R}^{d \times hd}$, where $h$ represents the head number, then there exists a quantum algorithm to prepare the A-Encoding state of $\frac{\partial C}{\partial F}$ and D-Encoding of $\frac{\partial C}{\partial X^{in}}$, where $F$ represents the parameters of the QAttn. The query complexity to the related D-Encodings is $\widetilde{O}(\frac{hd \log(n)}{\epsilon \delta^2})$.*

**Proof** *Firstly, we notice that*

$$\frac{\partial C}{\partial F} = \frac{\partial C}{\partial X^{out}} \frac{\partial X^{out}}{\partial F}, \tag{42}$$

*where $F$ contains $W$, $W_{vm}$, $W_{qm}$ and $W_{km}$ for $m = 0, 1, \cdots, h-1$. For each component of $F$, the expression of $\frac{\partial X^{out}}{\partial F}$ is based on Eq. (33), (35), (37), and (38). Therefore the A-Encoding of each component of $\frac{\partial X^{out}}{\partial F}$ can be prepared by A-Encoding or Block-encoding of $H$, $A'$, $K$, $Q$, and $V$. Subsequently, we construct the block-encoding of $\frac{\partial C}{\partial X^{out}}$ and apply this to the A-Encoding state $|\frac{\partial X^{out}}{\partial F}\rangle$, thereby preparing the A-Encoding of $\frac{\partial C}{\partial F}$. The query complexity to the related D-Encodings is $\widetilde{O}(\frac{hd}{\epsilon})$. Then we sample $|\frac{\partial C}{\partial F}\rangle$ $\widetilde{O}(\frac{\log(hd^2)}{\delta^2})$ times and obtain the sampled results. The query complexity to the related D-Encodings of this process is $\widetilde{O}(\frac{hd}{\epsilon \delta^2})$.*

*Secondly, we consider the derivative of the cost function relative to $X^{in}$:*

$$\frac{\partial C}{\partial X^{in}} = \frac{\partial C}{\partial X^{out}} \frac{\partial X^{out}}{\partial X^{in}} = \frac{\partial C}{\partial X^{out}} \frac{\partial X^{out}}{\partial H} \frac{\partial H}{\partial X^{in}}. \tag{43}$$

*The D-Encoding process of $\frac{\partial C}{\partial X^{in}}$ is detailed in the earlier segment of this section, with the query complexity of this process being $\widetilde{O}(hd^2/\epsilon)$. Following this, we sample $|\frac{\partial C}{\partial X^{in}}\rangle$ $\widetilde{O}(\frac{\log(dn)}{\delta^2})$ times, and build the D-Encoding of $\frac{\partial C}{\partial X^{in}}$ using the sampled results. The related D-Encodings' query complexity in this case is $\widetilde{O}(\frac{hd \log(n)}{\epsilon \delta^2})$.*

*In summary, the query complexity to the related D-Encodings of backpropagation is $\widetilde{O}(\frac{hd \log(n)}{\epsilon \delta^2})$.*

### D.5 QADD

The QAdd layer is written as:

$$X = X^{(1)} + X^{(2)}, \tag{44}$$

where $X, X^{(1)}, X^{(2)} \in \mathbb{R}^{d \times n}$. The forward pass and backpropagation of the QAdd layer are introduced as follows.

#### D.5.1 FORWARD

The input is the D-Encoding of $X^{(1)}$ and $X^{(2)}$, the output is the D-Encoding of $X$. We have

$$x_i = x_i^{(1)} + x_i^{(2)}, i = 0, 1, 2, \cdots, n-1, \tag{45}$$

therefore, the D-Encoding of $X$ is directly built by querying the D-Encoding of $X^{(1)}$ and $X^{(2)}$ once.

### D.5.2 BACKPROPAGATION

The input consists of the D-Encoding of $\frac{\partial C}{\partial X^{(\alpha)}}$ and $\frac{\partial C}{\partial X^{(\beta)}}$, where $\frac{\partial C}{\partial X^{(\alpha)}}$ is backpropagated from the next QAdd layer, and $\frac{\partial C}{\partial X^{(\beta)}}$ originates from the subsequent QNorm layer. The output comprises the D-Encoding of $\frac{\partial C}{\partial X^{(1)}}$ and $\frac{\partial C}{\partial X^{(2)}}$, with $\frac{\partial C}{\partial X^{(1)}}$ being directed backpropagated to the preceding QAdd layer, and $\frac{\partial C}{\partial X^{(2)}}$ being backpropagated to the previous layer.

We have

$$\frac{\partial C}{\partial X^{(1)}} = \frac{\partial C}{\partial X^{(2)}} = \frac{\partial C}{\partial X^{(\alpha)}} + \frac{\partial C}{\partial X^{(\beta)}}. \tag{46}$$

Therefore, the D-Encoding of $\frac{\partial C}{\partial X^{(1)}}$ and $\frac{\partial C}{\partial X^{(2)}}$ is built directly by querying the D-Encoding of $\frac{\partial C}{\partial X^{(\alpha)}}$ and $\frac{\partial C}{\partial X^{(\beta)}}$ once.

### D.6 QFFN

The feedforward layer is written as

$$X^{out} = W_2 f(W_1 X^{in} + b_1) + b_2, \tag{47}$$

where $f$ is the activation function. In our model, we employ the ReLU function as $f$. The forward pass and backpropagation of the QFFN layer are introduced as follows.

#### D.6.1 FORWARD PASS

The process involves the D-Encoding of the input matrix $X^{in}$ and subsequently produces the D-Encoding of the output matrix $X^{out}$. Each output element $x_i^{out}$ is determined through the equation:

$$x_i^{out} = W_2 f(W_1 x_i^{in} + b_1) + b_2, \quad i = 0, 1, 2, \cdots, n-1, \tag{48}$$

where each $x_i^{in}$ is a vector in $\mathbb{R}^d$. The D-Encoding of $X^{out}$ is then constructed by querying the D-Encoding of $X^{in}$ $d$ times.

#### D.6.2 BACKPROPAGATION

The input is the D-Encoding of $\frac{\partial C}{\partial X^{out}}$, and the output comprises the D-Encoding of $\frac{\partial C}{\partial X^{in}}$ along with the sampled $\frac{\partial C}{\partial F}$, where $F$ denotes the parameters in the QFFN layer.

First, we have

$$\frac{\partial C}{\partial x_i^{in}} = \frac{\partial C}{\partial x_i^{out}} \frac{\partial x_i^{out}}{\partial x_i^{in}}, \quad i = 0, 1, \cdots, n-1, \tag{49}$$

therefore, the D-Encoding of $\frac{\partial C}{\partial X^{in}}$ is constructed by querying the D-Encoding of $\frac{\partial C}{\partial X^{out}}$ and $X^{in}$ $d$ times.

Next, we prepare the state $|\frac{\partial C}{\partial F}\rangle$. We define $X^{mid} = W_1 X^{in} + [b_1, b_1, \cdots, b_1]$, similarly to $\frac{\partial C}{\partial X^{in}}$, the D-Encoding of $\frac{\partial C}{\partial X^{mid}}$ can also be constructed by querying the D-Encoding of $\frac{\partial C}{\partial X^{out}}$ and $X^{in}$ $d$ times. $F$ consists of $W_1, b_1$ and $W_2, b_2$, we have

$$[\frac{\partial C}{\partial W_2}, \frac{\partial C}{\partial b_2}] = \frac{\partial C}{\partial X^{out}}[\frac{\partial X^{out}}{\partial W_2}, \frac{\partial X^{out}}{\partial b_2}], (\frac{\partial X^{out}}{\partial W_2})_{ijkl} = \delta_{ik} f(X^{mid})_{jl}^T, (\frac{\partial X^{out}}{\partial b_2})_{ijk} = \delta_{ik}, \tag{50}$$

$$[\frac{\partial C}{\partial W_1}, \frac{\partial C}{\partial b_1}] = \frac{\partial C}{\partial X^{mid}}[\frac{\partial X^{mid}}{\partial W_1}, \frac{\partial X^{mid}}{\partial b_1}], (\frac{\partial X^{mid}}{\partial W_1})_{ijkl} = \delta_{ik}(X^{in})_{jl}^T, (\frac{\partial X^{mid}}{\partial b_2})_{ijk} = \delta_{ik}. \tag{51}$$

We construct block-encoding of $\frac{\partial C}{\partial X^{out}}$ and $\frac{\partial C}{\partial X^{mid}}$ by Lemma C.5. From Eq. (50) and (51), we can also prepare A-Encoding $|\frac{\partial X^{out}}{\partial W_2}\rangle, |\frac{\partial X^{out}}{\partial b_2}\rangle, |\frac{\partial X^{mid}}{\partial W_1}\rangle$, and $|\frac{\partial X^{mid}}{\partial b_1}\rangle$. Then $|\frac{\partial C}{\partial W_1}\rangle, |\frac{\partial C}{\partial W_2}\rangle, |\frac{\partial C}{\partial b_1}\rangle$ and $|\frac{\partial C}{\partial b_2}\rangle$ can be prepared and obtain its sampled distribution with $l_\infty$ tomography algorithm.

### D.7 QHEAD

The head layer of ViT is written as

$$X^{out} = W x_0^{in} + b, \tag{52}$$

where $x_0^{in} \in \mathbb{R}^d$, $X^{out} \in \mathbb{R}^K$, and $K$ represents the class number. The forward pass and backpropagation of the QHead layer are introduced as follows.

### D.7.1 Forward Pass

The input is the D-Encoding of $X^{in}$, and the output is the sampled $X^{out}$. The D-Encoding of $X^{out}$ is constructed by querying the D-Encoding of $X^{in}$ $d$ times. Subsequently, we prepare the A-Encoding state $|X^{out}\rangle$ using QDAC and obtain the sampled $X^{out}$ through the $l_\infty$ tomography algorithm. Finally, the classification label of $X$ is derived from the sampled results.

### D.7.2 Backpropagation

The input is the D-Encoding of $\frac{\partial C}{\partial X^{out}}$, and the output includes the D-Encoding of $\frac{\partial C}{\partial X^{in}}$ along with the sampled $\frac{\partial C}{\partial F}$, where $F$ denotes the parameters in the QHead layer.

Notice that the QHead layer is a simplified version of the QFFN layer without hidden layers. Therefore, the backpropagation of the QHead layer can be directly implemented using the backpropagation of the QFFN layer.

## E    Proof of Theorems 3.1

**Proof** *1. Forward pass*

*The query complexity of the QViT increases linearly with the number of encoder layers. Here, we analyze the complexity of one encoder layer of the QViT.*

*First, the dependence of the query complexity of the QPos, QAdd, QNorm, QFFN, and QHead layers on $d$ is the same as in the classical case, and the dependence on $n$ is $O(1)$.*

*Second, by Lemma D.2, the query complexity of the $X$ in the QAttn is $\widetilde{O}(\frac{d\log(n)}{\epsilon\delta^2})$, where $\delta$ represents the tomography error (Notice that we omit the head number $h$ here). The query complexity of the parameters in the QAttn is the query complexity of the $X$ multiplied by the factor $d$, because in the process $Wx_i$, the parameter matrix $W$ has $O(d^2)$ elements, $x_i$ has $O(d)$ elements. Therefore, the query complexity of the QAttn is $\widetilde{O}(\frac{d^2\log(n)}{\epsilon\delta^2})$.*

*Third, the query complexity of the following QAdd, QNorm, and QFFN is $O(d^2)$.*

*Therefore, the query complexity of one QViT encoder layer is $O(\frac{d^2\log(n)}{\epsilon\delta^2})$. Finally, the query complexity of the QHead layer is $O(d)$.*

*In summary, for an $l$-layer QViT, the query complexity of the forward pass is $\widetilde{O}(\frac{ld^2\log(n)}{\epsilon\delta^2})$.*

*2. Backpropagation*

*In the QHead layer, the D-Encoding of $\frac{\partial C}{\partial X^{out}}$ is built by querying the results obtained in the forward pass, and the query complexity to D-Encoding of $\frac{\partial C}{\partial X^{out}}$ is $O(d)$.*

*Next, we analyze the complexity of a layer of the QViT encoder from back to front.*

> *(1) The first layer is the QAdd, as introduced in Appendix D.5.2, the query complexity to the D-Encoding of $\frac{\partial C}{\partial X^{out}}$ is $O(1)$.*
>
> *(2) In the QFFN layer, we build the D-Encoding of $\frac{\partial C}{\partial X^{in}}$ and obtain the sampled $\frac{\partial C}{\partial F}$, where $F$ represents the parameters of the QFFN layer. As introduced in Appendix D.6.2, the query complexity to build the D-Encoding of $\frac{\partial C}{\partial X^{in}}$ is $O(d^2)$ because each $\frac{\partial C}{\partial x_i^{in}}$ is computed by $O(d^2)$ elements of $X^{out}$ and the QFFN layer parameters, and the query complexity to prepare each component of $|\frac{\partial C}{\partial F}\rangle$ is $\tilde{O}(d^2)$. Then we sample $|\frac{\partial C}{\partial F}\rangle$ $\widetilde{O}(\frac{\log(d^2)}{\delta^2})$ times and obtain the sampled results. The query complexity of the QFFN layer is $\widetilde{O}(\frac{d^2}{\delta^2})$.*
>
> *(3) In the QNorm layer, the query complexity to the D-Encoding of $\frac{\partial C}{\partial X^{out}}$ and $X^{in}$ is $O(d)$.*
>
> *(4) In the next QAdd layer, the query complexity to the D-Encoding of $\frac{\partial C}{\partial X^{out}}$ is $O(1)$.*

(5) By Lemma D.3, the query complexity of the D-Encodings of $X^{in}$, $X^{out}$, and $\frac{\partial C}{\partial X^{out}}$ in the QAttn layer is $\widetilde{O}(\frac{d \log(n)}{\epsilon \delta^2})$, and the query complexity of the parameters is $\widetilde{O}(\frac{d \log(n)}{\epsilon \delta^2})$ times the factor $d$. Therefore, the query complexity of the QAttn layer is $\widetilde{O}(\frac{d^2 \log(n)}{\epsilon \delta^2})$.

(6) In the next norm layer, the query complexity to the D-Encoding of $\frac{\partial C}{\partial X^{out}}$ and $X^{in}$ is $O(d)$.

We have analyzed the complexity of a QViT encoder layer, the query complexity is mainly determined by the QAttn layer, which is $\widetilde{O}(\frac{d^2 \log(n)}{\epsilon \delta^2})$.

Finally, in the QPos layer, the query complexity is $\widetilde{O}(\frac{\log(dn)}{\delta^2})$.

In summary, for an $l$-layer QViT, the query complexity of the backpropagation process is $\widetilde{O}(\frac{ld^2 \log(n)}{\epsilon \delta^2})$.

# F    ADDITIONAL RESULTS

The numerical results referred to in Section 2.2 that examine the cosine similarity between the output and input is shown in Figure 11.

In Section 4, we present the fine-tuning convergence curve of the CUB-200-2011 dataset, depicted in Figure 4. Additionally, we conduct tests on the Cifar-10, and Cifar-100 in Figure 12, yielding results similar to those obtained in the main text.

We conclude all the numerical results of the fine-tuning QViT models in Table 5, where each data represents the mean of five individual training processes.

Table 5: Classification accuracy (in %) of the QViT with multiple/single-reuse strategy. $\delta = 0$ represents the results of the classical vision transformer. Each data point is the average of five experimental results, with each experiment using a different random seed.

| Dataset | $\epsilon$ | Method | $\delta = 0$ | $\delta = 0.005$ | $\delta = 0.01$ | $\delta = 0.015$ | $\delta = 0.02$ |
|---|---|---|---|---|---|---|---|
| Cifar-10 | 0.000 | Multiple | 98.19 | 98.18 | 97.75 | 96.79 | 94.05 |
| | | Single | | 96.77 | 67.48 | 32.87 | 24.85 |
| | 0.004 | Multiple | 95.74 | 95.28 | 92.75 | 86.04 | 69.80 |
| | | Single | | 89.69 | 53.09 | 33.50 | 23.20 |
| | 0.008 | Multiple | 47.41 | 46.01 | 46.79 | 44.75 | 42.85 |
| | | Single | | 41.15 | 30.95 | 22.00 | 19.88 |
| Cifar-100 | 0.000 | Multiple | 89.41 | 89.23 | 87.50 | 81.69 | 56.21 |
| | | Single | | 87.44 | 27.43 | 10.43 | 6.61 |
| | 0.004 | Multiple | 74.13 | 70.30 | 56.68 | 40.99 | 29.02 |
| | | Single | | 48.24 | 18.16 | 9.01 | 4.89 |
| | 0.008 | Multiple | 14.13 | 14.59 | 13.87 | 13.36 | 12.59 |
| | | Single | | 12.20 | 8.09 | 4.01 | 1.84 |
| CUB-200-2011 | 0.000 | Multiple | 89.48 | 89.06 | 87.51 | 82.85 | 66.53 |
| | | Single | | 87.60 | 52.13 | 10.80 | 3.29 |
| | 0.004 | Multiple | 80.07 | 78.28 | 70.89 | 52.63 | 36.60 |
| | | Single | | 64.02 | 18.72 | 5.12 | 1.41 |
| | 0.008 | Multiple | 11.32 | 11.72 | 11.87 | 9.48 | 7.83 |
| | | Single | | 8.19 | 2.65 | 0.92 | 0.69 |
| Oxford III-T PETS | 0.000 | Multiple | 92.85 | 92.21 | 90.53 | 85.32 | 69.45 |
| | | Single | | 91.22 | 48.05 | 10.43 | 6.30 |
| | 0.004 | Multiple | 84.29 | 81.55 | 73.86 | 62.21 | 49.38 |
| | | Single | | 69.93 | 25.96 | 7.89 | 4.93 |
| | 0.008 | Multiple | 18.88 | 19.68 | 17.50 | 15.24 | 12.86 |
| | | Single | | 11.44 | 5.37 | 3.63 | 3.18 |

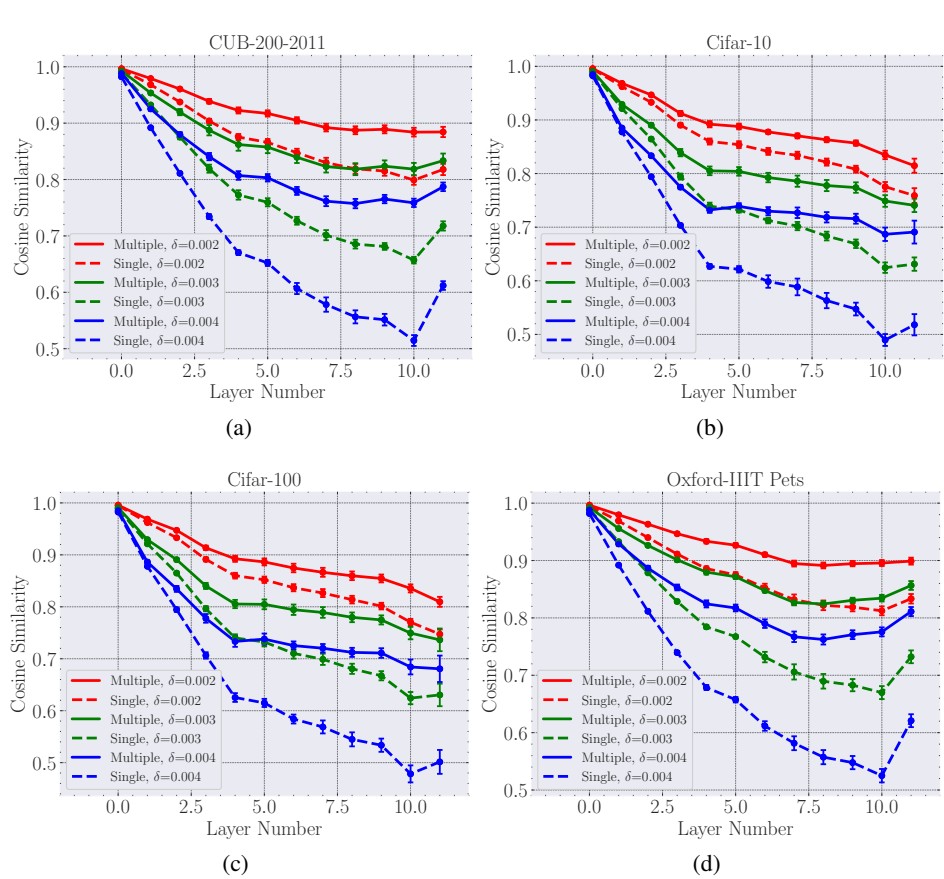

Figure 11: **The cosine similarity between the output of each layer of the QViT with multiple/single-reuse strategies and the correct output.** Subfigures (a), (b), (c), and (d) represent the CUB-200-2011, Cifar-10/100, and Oxford-IIIT Pets datasets respectively. The solid/dashed line represents the QViT with multiple/single-reuse strategy.

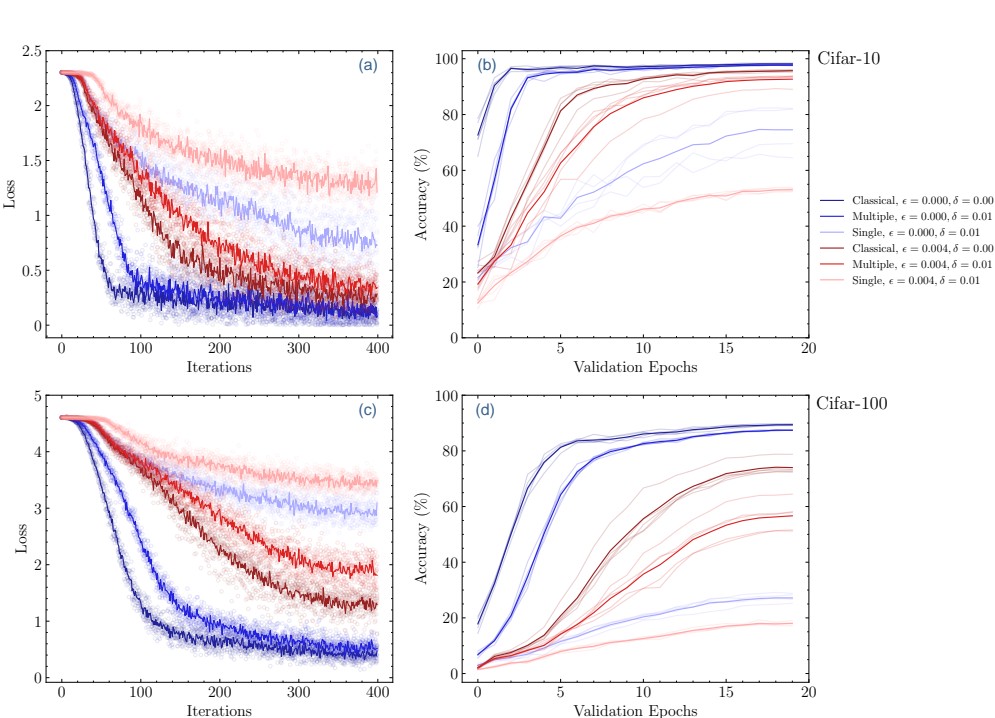

Figure 12: The fine-tuning convergence curve of Cifar-10 and Cifar-100 datasets.

