# OpenReview forum: "Measurement information multiple-reuse allows deeper quantum transformer"
_ICLR.cc/2025/Conference — ICLR 2025 Conference Withdrawn Submission_

### Official Review · Reviewer_gwnR · 2024-10-29

**Soundness:** 2
**Presentation:** 2
**Contribution:** 2
**Rating:** 3
**Confidence:** 3

**Summary:**

The paper introduces quantum vision transformers (QViTs), an implementation of vision transformers on fault-tolerant quantum computers, leveraging quantum tools for linear algebra. The implementation details follow somewhat Kerenidis et al [1], which introduces a similar scheme, but applying to quantum convolutional neural networks instead.

In addition to the implementation of QViTs, the authors propose a scheme called measurement information multiple-reuse (MIMR). In [1]  the measurement of one quantum layer is passed to the next quantum layer to allow for information sharing between quantum layers. This current work proposes that the measurement of all previous layers is provided as an input to all subsequent quantum layers.

The authors investigate the impact of MIMR on mitigating tomography errors across a range of benchmarks, investigating both the output of quantum layers vs baselines with no tomography error and also the overall accuracy of the resulting architectures.

[1] [Kerenidis et al, ICLR 2025](https://iclr.cc/virtual_2020/poster_Hygab1rKDS.html)

**Strengths:**

1. The implementation of MIMR is new, to my knowledge.
2. MIMR seems to provide performance benefits compared to single-information reuse.
3. The authors provide a detailed appendix including basics on quantum computing and discussions on implementation of individual layer forward and backward passes.

**Weaknesses:**

1. The query complexity presented in Theorem 3.1 can be better discussed and investigated. As written in the main text, the quantum query complexity goes as $O(ld^2\mathrm{polylog}(n)/\epsilon\delta^2)$ whereas classical goes as $O(lnd(n+d)\mathrm{log}(1/\epsilon))$. The text is quick to point out the quantum advantage in $n$, but does not fully discuss the quantum disadvantage at small values of $\epsilon$ and $\delta$. In practice, in remains unclear whether a QViT will be operating in the limit of large $n$, or small $\delta$ or $\epsilon$. These points are not discussed sufficiently nor investigated numerically. Consequently, the motivation for QViTs seems speculative at this stage.
2. The title, which states that MIMR allows for deeper quantum transformers, seems unjustified. Experiments are done with respect to different degrees of tomography errors, and do not vary depth.
3.  Experiments seem to only be conducted for a single random seed. It would be better to provide several runs of each experiment and present means and standard deviations or similar.
4. The benefit of MIMR is seemingly at very specific range of range of values: at $\delta=0$, single-reuse and multiple reuse are presented as identical. At $\delta=0.004$, multiple reuse performance also breaks down. It seems that the specific values at which MIMR might provide a benefit are $\delta \in \\{0.002, 0.003\\}$. Results for $\delta=0.001$ are not presented. This seems a particularly narrow range compared to the [work that establishes $l_\infty$ tomography](https://iclr.cc/virtual_2020/poster_Hygab1rKDS.html), which considers values of $\delta$ up to $0.1$.
5. The motivation of the work remains unclear. Since it relies on a host of quantum technologies not currently convincingly implemented (QRAM, amplitude encoding, $l_\infty$ tomography). Convincing error mitigation is currently not even demonstrated despite having been the subject of intense research. The techniques required seem even more involved: it seems quite plausible that transformers might not be of research interest by the time (if ever) this architecture is practically implementable on a real device.

**Questions:**

Suggestions:
1. Run experiments over multiple random seeds and report aggregated performance metrics.
2. An investigation over a wider range of $\delta$ would be beneficial to demonstrate the efficacy of MIMR. For example, [Kerenidis 2020](https://iclr.cc/virtual_2020/poster_Hygab1rKDS.html) considers values of $\delta=0.01$ and $\delta=0.1$. The range of values in this work seem small in comparison, and thus seem to be operating closer to a regime where the quantum advantage over classical in query complexity is diminished.
3. Provide a fuller discussion on where quantum might have an advantage over classical for QViTs, explaining clearly the tradeoffs between $\epsilon$, $\delta$ and $n$.
4. The long-term view of such quantum models should be made clear to ICLR readership, who are likely not quantum computing specialists. It is unclear when (or even if) tools like QRAM, tomography and amplitude encoding might be available on a practical device.
5. The idea of MIMR seems applicable to multiple architectures. Choosing to demonstrate it on a lightweight architecture and a simpler task seems to me to provide better opportunity to fully investigate its efficacy. By the time quantum computers can implement QViTs as presented, I think it's likely that the ML research community might have moved past transformers anyway. I appreciate this is a major change which is likely not implementable during the review period, but I do believe it would lead to a stronger and more general piece of work, in that it would allow MIMR to be presented as a general strategy as opposed to one pertaining to QViTs.
6. If stating that MIMR allows for deeper transformers, then there should be some numerical results with various transformer depths demonstrating the positive impact of MIMR vs single reuse.

---

> ### Author Response · Authors · 2024-11-27
>
> # Comment
> We appreciate the detailed feedback provided by the reviewers. We acknowledge the concerns and have addressed them comprehensively in our revised manuscript. Below, we respond specifically to the main points raised.
> # Discussion on error parameters
> We have conducted comprehensive numerical experiments in Section 4.3, fully considering the influence of  $ \delta $, $ \epsilon $, and the tomography strategy. These experiments were repeated five times to examine numerical stability and ensure the reliability of the results. We have expanded the parameter range and corrected a programming error identified during the execution of $ l_{\infty} $ tomography. The revised results now include plots explicitly showing the relationship between model performance and tomography
> and computing errors. These plots demonstrate that the "multiple reuse" strategy maintains its advantage up to the point where the model fails.
> # Numerical Results with Transformer Depths
> We acknowledge that the numerical experiments primarily focus on the impact of tomography errors rather than varying the transformer depths. the theoretical basis for our claim lies in the ability of the "multiple reuse" strategy to mitigate information loss, which is critical for constructing deeper networks, as discussed in Section 2. We will consider including numerical experiments varying transformer depths in future work to further substantiate this claim.
> # The Use of a Single Random Seed
> In the revised manuscript, we have repeated all experiments five times and reported the mean and standard deviation to ensure robustness and account for randomness. This change addresses the concern about numerical stability and provides a clearer picture of the results' reliability.
> # Scope of the "Multiple Reuse" Strategy
> We appreciate the suggestion to demonstrate the "multiple reuse" strategy on a lightweight architecture and simpler tasks. While this would provide a broader validation of its general applicability, we chose to focus on transformers due to their widespread use and have addressed some unique challenges with the nonlinear computations in quantum implementations. Nonetheless, we think that the "multiple reuse" strategy can be generalized to other architectures, and we plan to explore this direction in future work.
> # Long-Term Applicability and Practical Implementation Challenge
> We acknowledge that several quantum technologies required by our approach—such as QRAM, amplitude encoding, and $ l_\infty $ tomography—are not yet practically accessible. We have highlighted these limitations in the revised manuscript's Limitations section. While it is uncertain when fault-tolerant quantum computers will become available, we believe that the techniques we propose for linear computation and intermediate measurement optimization are valuable and could be applied to other frameworks beyond QViTs.

---

> > ### Comment · Reviewer_gwnR · 2024-11-28
> >
> > Thank you authors for your comments. Unfortunately my principal concerns have not been addressed:
> >
> > - The paper's main message still feels misleading. The argument that the information loss reduction associated with MIMR and that this allows deeper transformers is not demonstrated experimentally or theoretically beyond a heuristic argument made by the authors.
> > - I appreciate that the authors have increased the parameter range of experiments. But the performance of QViTs in general is dropping quite quickly as errors increase. The benefit of MIMR is there, but the performance of MIMR also drops quite quickly with  errors.
> > - It remains unclear whether the computational complexity of QViTs even has an advantage over classical transformers if a given accuracy is required given the inverse quadratic dependence on tomography error given in Theorem 3.1. The motivation to use QViTs over classical transformers needs better justification.

---

### Official Review · Reviewer_itQs · 2024-11-02

**Soundness:** 3
**Presentation:** 3
**Contribution:** 2
**Rating:** 5
**Confidence:** 4

**Summary:**

The authors provide a multiple-resued mechanism to mimic the residual connections. Based on this, the author constructed a Vision Transformer model with large parameters and representation power. The model's performance shows that training a large and deep quantum neural network becomes possible using this technique.

**Strengths:**

The paper is well written in narrative. The diagrams are clear and the background introduction is concise.

**Weaknesses:**

A review of the current method is lacking. In recent years, the research about quantum transformers and the mimicking of residual connections in quantum neural networks have been boosted. For example:
https://arxiv.org/abs/2406.04305
https://arxiv.org/abs/2209.08167
https://researchportal.hbku.edu.qa/en/publications/resqnets-a-residual-approach-for-mitigating-barren-plateaus-in-qu

Therefore, the novelty of this work is subtle.

**Questions:**

1. Please add some related work about residual connections in quantum neural networks.

2. Could you explain the difference between the "Multiple reuse" strategy with the available quantum residual connection implementation?

---

> ### Author Response · Authors · 2024-11-27
>
> # Comment
> We appreciate the detailed feedback provided by the reviewers. We acknowledge the concerns and have addressed them comprehensively in our revised manuscript. Below, we respond specifically to the main points raised.
> # Riview of current methods
> We have provided a review of current methods in the third paragraph of the Introduction, covering approaches based on variational networks and quantum linear algebra. These references include an overview of related works in the context of quantum transformers and residual connections in quantum neural networks.
> # Difference between the 'Multiple reuse' strategy with the available quantum residual connection.
> While residual connections have been implemented in quantum neural networks, our proposed "Multiple Reuse" strategy is distinct. It leverages quantum arithmetic operations and a ladder quantum circuit design to enhance model performance. Unlike conventional quantum residual connections, which intuitively involve adding up the amplitudes of quantum states (e.g., arXiv:2401.15871), our approach integrates additional computational structures to ensure better propagation of information and improved training efficiency.

---

> > ### Comment · Reviewer_itQs · 2024-11-28
> >
> > Thanks for the authors' comments. However, despite clarifying the distinction of the residual mechanism, the novelty of the manuscript technique is still limited. Meanwhile, as indicated by another reviewer, the main benefit of the "Multiple reuses" mechanism is rather intuitive, the author should consider demonstrating it quantitatively corresponding to its significance to this manuscript. Meanwhile, in architecture, the author didn't specify its distinction from the available quantum transformer, except for the "Multiple reuses", therefore, I will keep the current scores.

---

### Official Review · Reviewer_NUq7 · 2024-11-02

**Soundness:** 2
**Presentation:** 2
**Contribution:** 1
**Rating:** 3
**Confidence:** 5

**Summary:**

This paper proposes a measurement information multiple-reuse (MIMR) strategy to enhance quantum transformer models, specifically developing a quantum vision transformer (QViT). The MIMR method addresses limitations in quantum deep neural networks by reusing intermediate measurement information across layers, which mitigates information loss and improves efficiency in both forward pass and backpropagation.

**Strengths:**

1. The introduction of the MIMR scheme addresses a key bottleneck in quantum deep learning—information loss due to limited intermediate measurements. By reusing measurement information, the paper offers a fresh solution to build deeper quantum neural networks.

2. This work did a lot of experiments in image classification. It shows their method's effectiveness in such a task. This practical focus adds value to the proposed method, as it directly showcases the model's utility.

**Weaknesses:**

1. In this paper, the authors state that they are proposing a quantum transformer. However, as far as I understand, what they are proposing is a hybrid quantum-classical transformer. As Figure 2 shows, the method keeps converting the tensor between quantum and classical states. Therefore, authors may make this more clear and precise.

2. The method keeps converting the tensor between quantum and classical states. Will such conversions be time-consuming?

3. In this work, the authors did not implement their method on a real quantum computer. There are a lot of quantum computers available for use, such as the IBM Quantum Cloud Platform (IBM-Q). It is not clear how the authors implemented QVit. It looks like they are just doing the simulation instead of running the method on the actual quantum machine, but it is not clear how the simulation is implemented.

4. In quantum computing, the quantum noise is a very critical problem. However, there is no discussion about the quantum noise. This problem is significant when converting the tensors from a quantum state to a classical state.

5. Authors use pre-trained QViT, but it is not clear which dataset was used for pre-trianing.

6. The application is limited. Only image classification is discussed. Since it is about the transformers, the application in language processing is also important.

**Questions:**

1. Are you running the simulation? Can you please make it more clear that how do you implement the network? Have you used Qiskit or another quantum library?

2. The method keeps converting the tensor between quantum and classical states. Will such conversions be time-consuming?

3. How does the quantum noise affect this work? How do you reduce the effect of the quantum noise?

4. Which dataset is used for pertaining?

---

> ### Author Response · Authors · 2024-11-27
>
> # Comment
> We appreciate the detailed feedback provided by the reviewers. We acknowledge the concerns and have addressed them comprehensively in our revised manuscript. Below, we respond specifically to the main points raised.
> # Clarification of Quantum Transformer Implementation
> Our quantum transformer alternates between digital encoding and amplitude encoding during computations. Digital encoding computations are executed using quantum arithmetic operations, while amplitude encoding computations utilize quantum linear algebra. The definitions of digital encoding and amplitude encoding are provided in the main text. The execution of the quantum transformer includes quantum-classical data conversion following the quantum attention layer. Specifically, the amplitude-encoded state output from the attention layer is sampled and stored in QRAM. The QRAM is then queried to prepare operations in digital encoding form, and computations proceed in digital encoding until the next attention layer. This process is elaborated in Appendix D of the revised manuscript.
> # Performance Impact of Quantum-Classical Data Conversion
> We acknowledge that quantum-classical data conversion affects the performance of our proposed quantum transformer. To address this, we optimized the complexity of intermediate measurements in two ways:
> First, we employed the $l_{\infty}$ sampling algorithm, whose complexity scales logarithmically with the dimension $ n $ of the quantum state.
> Second, we introduced the Measurement Reuse strategy, which mitigates information loss caused by intermediate measurements. This enables the model to remain trainable even when sampling errors are relatively high. These optimizations reduce the overall impact of quantum-classical data conversion on complexity.
> # Implementation on a Real Quantum Computer:
> Our algorithm is designed for fault-tolerant quantum computers, which are not yet available for practical use. Therefore, it cannot currently be demonstrated on NISQ devices. We acknowledge this limitation and have added a discussion in the revised manuscript's final section. Additionally, details about our simulation implementation have been included in the updated text.
> # Quantum Noise Considerations
> Since our algorithm is intended for execution on fault-tolerant quantum computers, quantum error correction techniques in such systems theoretically suppress errors caused by quantum noise. Thus, we did not explicitly address quantum noise in this work. However, we did consider statistical errors introduced by intermediate measurements and computational precision errors in quantum arithmetic operations. The impact of these errors on the performance of the quantum transformer has been validated through numerical experiments.
> # Pre-trained Model Dataset
> The pre-trained QViT model was trained on the ImageNet-21k dataset. This information has been clarified in the revised manuscript.
> # Application Scope Beyond Image Classification
> Thank you for this valuable suggestion. In this work, we primarily focused on image classification. Expanding the application of our quantum transformer to natural language processing is a promising direction, and we plan to explore this in future work.

---

> > ### Comment · Reviewer_NUq7 · 2024-11-28
> >
> > Thank you for your rebuttal. However, the limited results remain a major concern. Therefore, I have to maintain my original rating.

---

### Official Review · Reviewer_UYAH · 2024-11-02

**Soundness:** 2
**Presentation:** 3
**Contribution:** 1
**Rating:** 5
**Confidence:** 4

**Summary:**

This paper presents a method for leveraging quantum circuits and their properties to enhance the performance of transformers on a quantum computer. The authors additionally propose utilizing mid-circuit measurements, which serve as a quantum analogue to deep residual connections, as a strategy to construct deeper quantum circuits.

**Strengths:**

1. Fair story and reasonable thought to replace traditional network module operations with quantum gates.
2. Use quantum gates to enhance computational efficiency within transformer architectures.
3. Leverage mid-circuit measurement information to construct deeper quantum transformers.

**Weaknesses:**

1. Lack of Novelty: The approach of replacing mathematical operations with quantum gates is not novel, as this concept has been widely explored and documented, notably in the qBLAS library. The paper does not present significant innovation beyond these existing approaches, which diminishes its contribution to the field.

2. Limited In-depth Consideration: While the authors propose using residual connections within quantum circuits—an idea inspired by classical neural network architectures—this adaptation requires more critical evaluation. In classical settings, residual connections support the training of deeper models; however, applying this concept to quantum circuits entails significant challenges. Specifically, implementing residual connections in quantum circuits would necessitate re-preparing the quantum states following mid-circuit measurements, which imposes a substantial computational burden. This could potentially negate the speed advantages typically offered by quantum computing. The primary challenge, as acknowledged in quantum computing literature, for training deep quantum circuits relates to the fact that deeper quantum circuits can approximate more easily Haar random quantum states.

Additionally, the concept of residual connections within quantum circuits has already been explored, as evidenced in [1]. The paper would benefit from a clearer discussion of how its approach differs from or builds upon prior work in this area.

[1] https://arxiv.org/abs/2401.15871

**Questions:**

Could you provide a more detailed theoretical explanation on how this approach might enable the training of deeper models within a quantum framework, despite the additional requirements for state re-preparation? This clarification would strengthen the justification for the proposed method.

**Details Of Ethics Concerns:**

No.

---

> ### Author Response · Authors · 2024-11-27
>
> # Comment
> We appreciate the detailed feedback provided by the reviewers. We acknowledge the concerns and have addressed them comprehensively in our revised manuscript. Below, we respond specifically to the main points raised.
> # Novelty of the Work
> First, we propose a Measurement Reuse strategy that effectively mitigates the issue of information loss caused by intermediate measurements. Second, the execution flow of our proposed quantum transformer goes beyond quantum linear algebra. We alternate between digital encoding and amplitude encoding during computations: the digital encoding calculations are performed using quantum arithmetic operations, while the amplitude encoding calculations leverage quantum linear algebra. This alternation optimizes the success rate of non-unitary operations executed via quantum linear algebra, which typically decreases exponentially with the number of layers. These innovations highlight the unique contributions of our work.
> # Addressing Residual Connections and Computational Challenges
> The core idea of our Measurement Reuse strategy ensures an uninterrupted information pathway throughout the model's execution, despite the presence of intermediate measurements. Residual connections, in our context, can be viewed as two pathways: intermediate measurements are applied to only one pathway, enabling multiple reuses of the measured information. Our approach is distinct from the method discussed in arxiv:2401.15871, which remains within the framework of variational quantum circuits and cannot be directly applied to our quantum transformer model. This distinction is now clarified in the revised manuscript.
> # Theoretical Explanation and Justification
> As outlined above, our Measurement Reuse strategy ensures that there is always an uninterrupted information pathway during the execution of the quantum transformer model. All intermediate measurement information can propagate through this pathway and be reused effectively. Figure 1 in the manuscript provides a visual illustration of this concept. By addressing the information loss issue caused by intermediate measurements as the model depth increases, our approach enables the training of deeper models. This directly supports the feasibility and advantages of our proposed quantum transformer in achieving deeper quantum model training.

---

> > ### Comment · Reviewer_UYAH · 2024-12-02
> >
> > I thank the authors for their reply. After carefully evaluating the changes, I am afraid I have to keep my original rating. The general strategy of measurement-reuse is to ensure the information flows uninterrupted. With the classical residual connection and the corresponding quantum version, I do not think it provides valuable insights which are significantly different from existing work. A general suggestion: when there are changes into the draft, please highlight it in different colors. Otherwise, it is difficult to track where the changes are.

---

### Note · Authors · 2024-12-08

I have read and agree with the venue's withdrawal policy on behalf of myself and my co-authors.